# High-dimensional limit theorems for SGD: Effective dynamics and critical scaling

**Gérard Ben Arous**
Department of Mathematics
Courant Institute
New York University
New York, NY
`benarous@cims.nyu.edu`

**Reza Gheissari**
Miller Institute
Departments of EECS and Statistics
University of California, Berkeley
Berkely, CA
`gheissari@berkeley.edu`

**Aukosh Jagannath**
Department of Statistics and Actuarial Science
Department of Applied Mathematics
University of Waterloo
Waterloo, ON `a.jagannath@uwaterloo.ca`

## Abstract

We study the scaling limits of stochastic gradient descent (SGD) with constant step-size in the high-dimensional regime. We prove limit theorems for the trajectories of summary statistics (i.e., finite-dimensional functions) of SGD as the dimension goes to infinity. Our approach allows one to choose the summary statistics that are tracked, the initialization, and the step-size. It yields both ballistic (ODE) and diffusive (SDE) limits, with the limit depending dramatically on the former choices. We find a critical scaling regime for the step-size below which this "effective dynamics" matches gradient flow for the population loss, but at which, a new correction term appears which changes the phase diagram. About the fixed points of this effective dynamics, the corresponding diffusive limits can be quite complex and even degenerate. We demonstrate our approach on popular examples including estimation for spiked matrix and tensor models and classification via two-layer networks for binary and XOR-type Gaussian mixture models. These examples exhibit surprising phenomena including multimodal timescales to convergence as well as convergence to sub-optimal solutions with probability bounded away from zero from random (e.g., Gaussian) initializations.

## 1 Introduction

**Background.** Stochastic gradient descent (SGD) is the go-to method for large-scale optimization problems in data science. It is used to train complex parametric models on high-dimensional data. Since its introduction in [39], there has been a tremendous amount of work in analyzing its evolution.

In fixed dimensions, the asymptotic theory of SGD, and stochastic approximations more broadly, is by now classical. There have been works on path-wise limit theorems, such as functional central limit theorems and even large deviations principles [39, 31, 28, 22, 17, 7, 16, 6]. At the core of this line of work is the idea that in the limit where the step-size, or learning rate, tends to zero, the trajectory of SGD with a fixed loss function (appropriately rescaled in time) converges to the solution of gradient flow for the population loss with the same initialization. Recently there has been considerable interest in quantifying the rate of this trajectory-wise convergence to higher order, in terms of a diffusion approximation. Namely, there are many works developing asymptotic expansions of the trajectory in

36th Conference on Neural Information Processing Systems (NeurIPS 2022).

the learning rate [29, 23, 25, 1, 26]. Motivated by this, there is a rich line of work bounding the time to equilibrium for the associated diffusion approximation (as well as Langevin–type modifications) under uniform ellipticity assumptions [29, 36, 11, 53]. There is also an interesting line of work obtaining PDE limits in the "shallow network" regime where the dimension of the parameter space diverges but the dimension of the data remains constant: see e.g., [32, 40, 12, 45, 2].

In recent years, there has been considerable interest in understanding the *high-dimensional setting*, where one is constrained in the amount of data or the run-time of the algorithm due to the high-dimensional nature of the data and the complexity of the model being trained. In these regimes, one cannot simply take the learning rate to be arbitrarily small as this would force an unlimited sample size and run-time. This is a common issue in high-dimensional statistics and the standard analytic approach is to study regimes where the sample size scales with the dimension of the problem [50, 51].

For SGD with constant learning rate, there has been recent progress on quantifying the dimension dependence of the sample complexity for various tasks on general (pseudo or quasi-) convex objectives [8, 9, 44, 34, 21, 15] and special classes of non-convex objectives [19, 47, 3]. There has also been important work on scaling limits as the dimension tends to infinity for the specific problems of linear regression [52, 35], Online PCA [52, 24], and phase retrieval [47] from random starts, and teacher-student networks [41, 42, 20, 49] and two-layer networks for XOR Gaussian mixtures [37] from warm starts. We also note that the study of high-dimensional regimes of gradient descent and Langevin dynamics have a history from the statistical physics perspective, e.g., in [13, 14, 43, 30, 10, 27].

**Our contributions.** We develop a unified approach to the scaling limits of SGD in high-dimensions with constant learning rate that allows us to understand a broad range of estimation tasks. One of course cannot develop a high-dimensional scaling limit for the full trajectory of SGD as the dimension of the underlying parameter space is growing. On the other hand, in practice, one is rarely interested in the full trajectory; instead one typically tracks the trajectory of various summary statistics of the algorithm's evolution, such as the loss, the amplitude of various weights, or correlations between the classifier and the ground truth (in a supervised setting). We show in Theorem 2.2 that under mild regularity assumptions, the evolution of these summary statistics converges as the dimension grows to the solution of a system of (possibly stochastic) differential equations. These *effective dynamics* depend dramatically on the initializations (warm vs. random or cold), the parameter regions in which one is developing the scaling limit, and the scaling of the step-size with the dimension.

In practice, SGD often exhibits two types of phases in training: *ballistic phases* where the summary statistics macroscopically change in value, and *diffusive phases*, where they fluctuate microscopically. (During training, the evolution can start with either, and can even alternate multiple times between these phases.) Our approach allows us to develop scaling limits for both types of phases.

In ballistic phases, the effective dynamics are given by an ordinary differential equation (ODE) and the finite-dimensional intuition that the summary statistics evolve under the gradient flow for the population loss is correct provided the (constant) learning rate is sufficiently small in the dimension. When the learning rate follows a certain *critical* scaling—matching scalings commonly used in the high-dimensional statistics literature—an additional correction term appears. At this critical scaling, the phase portrait deviates significantly from that of the population gradient flow. Furthermore, in microscopic neighborhoods of the fixed points of this ODE, the effective dynamics become diffusive and are given by SDEs which can exhibit a wide range of (possibly degenerate) behaviors. We note that the appearance of the correction term in the ballistic phase was first observed in the setting of teacher-student networks in [41, 42] and very recently investigated in detail in [49].

As a simple, first example of the departure of the effective dynamics in the critical step-size regime from the classical perspective, we study estimation for spiked matrix and tensor models in Section 3. In these models, the effective dynamics are exactly solvable and when the step-size scales critically with the dimension, in the ballistic phase the dynamics have additional fixed points as compared to the population gradient flow. The stability of these fixed points exhibit sharp transitions at special signal-to-noise ratios. When initialized randomly, the SGD starts in a microscopic neighborhood of an uninformative such fixed point, within which its effective dynamics become diffusive and exhibit a sharp transition between stable and unstable Ornstein–Uhlenbeck processes.

To demonstrate our approach on more complex classification tasks typically studied using neural networks, we study a Gaussian mixture model analogue of the classical XOR problem in Section 5. (The XOR problem is arguably the canonical example of a decision boundary requiring at least

two-layers to represent [33].) Here we find that the natural summary statistics are 22 dimensional, and their (ballistic) effective dynamics exhibit a rich phenomenology between some 39 fixed point regions of varying topological dimension. Surprisingly, we find that if we initialize the weights of the network randomly (following a Gaussian distribution), then the algorithm will converge to a classifier with macroscopic generalization error with probability $29/32$ and then follow a degenerate diffusion.

Before delving into the XOR problem, we first analyze the classification of a two component Gaussian mixture model in Section 4. This task is of course best solved using a one-layer network i.e., logistic regression, but with a two-layer network it exhibits some similar phenomenologies to the XOR problem while being more amenable to finer analysis. Here, we again find that if with random initial weights, with probability $1/2$ the SGD will first converge to a classifier with macroscopic generalization error, and then follow a degenerate diffusion in a microscopic neighborhood of that set of unstable fixed points. We demonstrate this both empirically for positive signal-to-noise ratio and theoretically in the limit where the SNR tends to zero after the dimension tends to infinity.

## 2   Main result

Suppose that we are given a sequence of i.i.d. data $Y_1, Y_2, \ldots$ taking values in $\mathcal{Y}_n \subseteq \mathbb{R}^{d_n}$ with law $P_n \in \mathcal{M}_1(\mathbb{R}^{d_n})$, and a loss function $L_n : \mathcal{X}_n \times \mathcal{Y}_n \to \mathbb{R}$, where here $\mathcal{X}_n \subseteq \mathbb{R}^{p_n}$ is the parameter space. Consider online stochastic gradient descent with constant learning rate, $\delta_n$, which is given by

$$X_\ell = X_{\ell-1} - \delta_n \nabla L_n(X_{\ell-1}, Y_\ell),$$

with possibly random initialization $X_0 \sim \mu_n \in \mathcal{M}_1(\mathcal{X}_n)$. Our interest is in understanding this evolution, $(X_\ell)$, in the regime where both $p_n$ and $d_n \to \infty$ as $n \to \infty$. To this end, suppose that there is a finite collection of summary statistics of $(X_\ell)$ whose evolution we are interested in. More precisely, suppose that we are given a sequence of functions $\mathbf{u}_n \in C^1(\mathbb{R}^{p_n}; \mathbb{R}^k)$ for some fixed $k$, where $\mathbf{u}_n(x) = (u_1^n(x), ..., u_k^n(x))$, and our goal is to understand the evolution of $\mathbf{u}_n(X_\ell)$.

To develop a scaling limit, we need some assumptions on the relationship between how the step-size scales in relation to the loss, its gradients, and the data distribution. To this end let $H(x, Y) = L_n(x, Y) - \Phi(x)$, where $\Phi(x) = \mathbb{E}[L_n(x, Y)]$. Throughout the following, we suppress the dependence of $H$ on $Y$ and simply write $H(x)$, and instead view $H$ as a random function of $x$.

**Definition 2.1.** We say that a triple $(\mathbf{u}_n, L_n, P_n)$ is $\delta_n$-**localizable** if there is an exhaustion by compact sets of $\mathbb{R}^k$, call it $(E_K)_K$, and constants $0 < C(K) < \infty$ (independent of $n$) such that

1. $\max_{1 \le i \le k} \sup_{x \in \mathbf{u}_n^{-1}(E_K)} \|\nabla^j u_i\|_{op} \le C(K) \cdot \delta_n^{-(3-j)/2}$ for $j = 2, 3$;

2. $\sup_{x \in \mathbf{u}_n^{-1}(E_K)} \|\nabla \Phi\| \le C(K)$, and

3. $\max_{1 \le i \le k} \sup_{x \in \mathbf{u}_n^{-1}(E_K)} \mathbb{E}[\langle \nabla H, \nabla u_i \rangle^4] \le C(K) \delta_n^{-2}$, and

   $\max_{1 \le i \le k} \sup_{x \in \mathbf{u}_n^{-1}(E_K)} \mathbb{E}[\langle \nabla^2 u_i, \nabla H \otimes \nabla H - V \rangle^2] = o(\delta_n^{-1}).$

When these hold we call the sequence $(E_K)$ the **localizing sequence** of $(\mathbf{u}_n, L_n, P_n)$.

Localizability is wider than uniform Lipchitz or smoothness assumptions common to the literature. In particular, it does not imply that the population loss is Lipschitz everywhere, as we may have that $\bigcup_K \mathbf{u}_n^{-1}(E_K)$ does not cover $\mathbb{R}^{p_n}$, nor does it imply uniform smoothness of $L$ as we will be taking $\delta_n \to 0$ with $n$. To motivate the scaling relations between 2–3, note that if $\delta_n \asymp p_n^{-1}$, corresponding to linear sample complexity, the scaling relation is the same as what one would get e.g., if $\nabla H$ were a random vector with independent entries of bounded variance in $\mathbb{R}^{p_n}$ and e.g., $\nabla u_i = \delta_n^{-1/2} e_1$.

We now turn to the statement of our main result. Let $\mathcal{P}_k$ denote the space of positive semi-definite $k \times k$ matrices and for a function $f$ and measure $\mu$ we let $f_* \mu$ denote the push-forward of $\mu$. Let $J_n = (\nabla u_\ell)$ denote the Jacobian of the summary statistics. Also, let $V(x) = \mathbb{E}[\nabla H(x) \otimes \nabla H(x)]$ denote the covariance matrix for $\nabla H$ at a point $x$ and define the corresponding first and second-order differential operators,

$$\mathcal{A}_n = \sum \partial_i \Phi \partial_i, \qquad \text{and} \qquad \mathcal{L}_n = \frac{1}{2} \sum V_{ij} \partial_i \partial_j.$$

We then have the following convergence result.

**Theorem 2.2.** *Let $(X_\ell^{\delta_n})_\ell$ be the SGD initialized from $X_0 \sim \mu_n$ for $\mu_n \in \mathcal{M}_1(\mathbb{R}^{p_n})$ with learning rate $\delta_n$ for the loss $L_n(\cdot,\cdot)$ and data distribution $P_n$. Suppose that $\mathbf{u}_n$ is such that the triple $(\mathbf{u}_n, L_n, P_n)$ is $\delta_n$-localizable with localizing sequence $(E_K)$. Suppose furthermore that there exists locally Lipschitz $\mathbf{h} : \mathbb{R}^k \to \mathbb{R}^k$ and $\Sigma : \mathbb{R}^k \to \mathcal{P}_k$, such that for every $K$,*

$$\sup \|(-\mathcal{A}_n + \delta_n \mathcal{L}_n)\mathbf{u}_n(x) - \mathbf{h}(\mathbf{u}_n(x))\| \to 0\,, \tag{2.1}$$

$$\sup \|\delta_n J_n V J_n^T - \Sigma(\mathbf{u}_n(x))\| \to 0\,, \tag{2.2}$$

*where the suprema are over $x \in \mathbf{u}_n^{-1}(E_K)$. Then if we let $(\mathbf{u}_n(t))_t$ be the linear interpolation of $(\mathbf{u}_n(X_{\lfloor t\delta_n^{-1}\rfloor}^{\delta_n}))_t$, and the initial data $\mu_n$, is such that $(\mathbf{u}_n)_*\mu_n \to \nu$ weakly, then, $(\mathbf{u}_n(t))_t \to (\mathbf{u}_t)_t$ weakly as $n \to \infty$, where $\mathbf{u}_t$ is the solution to*

$$d\mathbf{u}_t = \mathbf{h}(\mathbf{u}_t)dt + \sqrt{\Sigma(\mathbf{u}_t)}d\mathbf{B}_t\,, \tag{2.3}$$

*initialized from $\nu$, where $\mathbf{B}_t$ is a standard Brownian motion in $\mathbb{R}^k$.*

Theorem 2.2 is proved in Appendix C. The proof can be seen as a version of the martingale problem (see [46]) for high-dimensional SGD.

## 2.1 Comparison to fixed dimensional perspective: critical v.s. subcritical step-sizes

Let us compare this with the classical limit theory of SGD in fixed dimension. For the sake of this discussion, suppose that not only does (2.1) hold, but each of the two terms $\mathcal{A}_n\mathbf{u}$ and $\delta_n\mathcal{L}_n\mathbf{u}$ individually admit $n \to \infty$ limits: namely that there exists $\mathbf{f}, \mathbf{g} : \mathbb{R}^k \to \mathbb{R}^k$ such that

$$\sup_{x \in \mathbf{u}_n^{-1}(E_K)} \|\mathcal{A}_n\mathbf{u}_n(x) - \mathbf{f}(\mathbf{u}_n(x))\| \vee \|\delta_n\mathcal{L}_n\mathbf{u}_n(x) - \mathbf{g}(\mathbf{u}_n(x))\| \to 0\,, \tag{2.4}$$

in which case, evidently (2.1) holds with $\mathbf{h} = -\mathbf{f} + \mathbf{g}$. When (2.4) both hold, we call $\mathbf{f}, \mathbf{g}$ and $\Sigma$ the **population drift**, the **population corrector**, and the **diffusion matrix** of $\mathbf{u}$ respectively.

From the fixed dimensional perspective, when (2.4) holds, one predicts $\mathbf{u}$ to asymptotically solve

$$d\mathbf{u}_t = -\mathbf{f}(\mathbf{u}_t)dt\,, \tag{2.5}$$

with initial data $\mathbf{u}_0 \sim \mathbf{u}_*\mu$. as this is its evolution under gradient descent on the population loss $\Phi$. Evidently this perspective only applies in the high-dimensional limit of Theorem 2.2 if both the population corrector $\mathbf{g}$ and the diffusion matrix $\Sigma$ are zero. We find that for any triple $(\mathbf{u}_n, L_n, P_n)$, there is a scaling of the learning rate $\delta_n$ with $n$ below which $\mathbf{g} = \Sigma = 0$, and the effective dynamics agree with the population dynamics (2.5)—call this the **sub-critical** scaling regime, where the classical perspective applies—and a **critical** scaling regime in which $\mathbf{g}$ and $\Sigma$ may be non-zero, and the high-dimensionality induces non-trivial corrections to $\mathbf{f}$. (In the case of teacher–student networks, the terms $\mathbf{f}$ and $\mathbf{g}$ can be compared to the "learning" and "variance" terms in Eq. (14a) of [49].)

To see this, notice that if the triple $(\mathbf{u}_n, L_n, P_n)$ is $\delta_n$-localizable for some $\delta_n \to 0$, then it is also $\delta_n'$-localizable for every sequence $\delta_n' = O(\delta_n)$. If furthermore (2.4)-(2.2) hold for $\delta_n$ with some $\mathbf{f}, \mathbf{g}$ and $\Sigma$, then these limits also exists for $\delta_n' = o(\delta_n)$ with the same $\mathbf{f}$ but with $\mathbf{g} = \Sigma = 0$. As such, there can be exactly one scaling of $\delta_n$ with $n$ at which $\mathbf{g}$ or $\Sigma$ may be non-zero, and for all smaller scales of learning rate, the fixed-dimensional perspective of (2.5) applies.[1]

## 2.2 Ballistic vs. diffusive behavior of effective dynamics

In all of our examples, the diffusion matrix for the effective dynamics of the most natural choice of summary statistics is zero even in the critical scaling regime where $\mathbf{h} \neq \mathbf{f}$. We call this the **ballistic limit**. In this case, the effective dynamics of the summary statistics is given by the ODE system

$$d\mathbf{u}_t = \mathbf{h}(\mathbf{u}_t)dt\,. \tag{2.6}$$

In these settings, the phase portrait of the summary statistics is asymptotically that of this flow.

By construction of the scaling limit, the phase portrait of the ballistic limit only describes the evolution of summary statistics on length-scales that are order 1 and time-scales that are order $1/\delta_n$. If one is

---

[1]Note that if $\delta_n = o(\delta_n')$, then limiting $\mathbf{g}, \Sigma$ may not exist for $\delta_n'$, so there is no super-critical regime.

then interested in the evolution of $\mathbf{u}_n$ in microscopic $o(1)$ neighborhoods of the fixed points of (2.6), Theorem 2.2 also allows one to develop separate **diffusive limits** there.

To understand diffusive regimes, one must apply Theorem 2.2 to a re-centered and re-scaled version of the summary parameters, $\tilde{\mathbf{u}}_n(t) = \delta_n^{-\alpha}(\mathbf{u}_n(t) - \mathbf{u}_\star)$ where $\mathbf{u}_\star$ is a fixed point of (2.6).[2] To apply Theorem 2.2, $\alpha$ must be chosen appropriately so that the triple $(\tilde{\mathbf{u}}_n(t), L_n, P_n)$ is $\delta_n$-localizable and to pick out the next order drifts for $\tilde{u}$—the first order term being zero microscopically close to $\mathbf{u}_\star$—and such that the initial data still converges $(\tilde{\mathbf{u}}_n)_*\mu_n \to \tilde\nu$.

This then leads to the **rescaled effective dynamics** of the summary statistics $\mathbf{u}_n$ near $\mathbf{u}_\star$:

$$d\tilde{\mathbf{u}}_t = \tilde{\mathbf{h}}(\tilde{\mathbf{u}}_t)dt + \tilde\Sigma^{1/2}(\tilde{\mathbf{u}}_t)d\mathbf{B}_t, \qquad \text{with} \qquad \tilde{\mathbf{u}}_0 \sim \tilde\nu. \tag{2.7}$$

The rescaled effective dynamics are similar in spirit to diffusion approximations typically one finds for the evolution of SGD near critical points in fixed dimensions. However, we note two important differences as compared to this perspective. Firstly, since this is a high-dimensional limit of general summary statistics, (2.7) applies in a neighborhood of a fixed point of the effective ODE system (2.6), rather than the population dynamics (2.5). Secondly, in many examples (indeed all the ones we study) the SDE's we get are degenerate to some degree, so that uniform ellipticity assumptions typically used to understand hitting and mixing times in these regimes do not apply.

## 3    Matrix and Tensor PCA

As our first example, we consider the problems of spiked matrix models and spiked tensor models [38] using SGD. These examples are exactly solvable and only require two summary statistics, a correlation observable and a radial term. Even with this relative simplicity, we encounter a wide range of ODE and SDE limits. Interestingly, by means of these SDE limits, we can sharply identify the signal-to-noise thresholds for solving the recovery problem by means of the SGD.

Suppose that we are given data of the form $Y = \lambda v^{\otimes k} + W$ where $W$ is an i.i.d. Gaussian $k$-tensor, $v \in \mathbb{R}^n$ is a unit vector, and $\lambda = \lambda_n > 0$ is the signal-to-noise ratio. Our goal is infer $v$. We take as loss the (negative) log-likelihood namely, $L(x, Y) = ||Y - x^{\otimes k}||^2$.[3] The pair of summary statistics $m = m(x) := \langle x, v \rangle$ and $r_\perp^2 = r_\perp^2(x) := \|x - mv\|^2 = \|x\|^2 - m^2$ are such that $\Phi(x) = -2\lambda m^k + (r_\perp^2 + m^2)^k + c$, and the law of $L$ only depends on them: see Section D.1.

For the pair $\mathbf{u}_n = (u_1, u_2) = (m, r_\perp^2)$, Theorem 2.2 yields the following effective dynamics. In our normalization with $\lambda > 0$ fixed, the regime $\delta_n = o(1/n)$ is sub-critical and the regime $\delta_n = \Theta(1/n)$ is critical; we focus on this normalization for presentation, but note that with different scalings of $\lambda_n$, the critical learning rates change. For notational simplicity, let $R^2 := m^2 + r_\perp^2$.

**Proposition 3.1.** *Fix $k \geq 2$, $\lambda > 0$, $c_\delta > 0$ and let $\delta_n = c_\delta/n$.[4] Then $\mathbf{u}_n(t)$ converges as $n \to \infty$ to the solution of the following ODE initialized from $\lim_{n\to\infty}(\mathbf{u}_n)_*\mu_n$:*

$$\begin{aligned}
\dot{u}_1 &= 2u_1(\lambda k u_1^{k-2} - kR^{2k-2}), \\
\dot{u}_2 &= -4kR^{2(k-1)}(u_2 - c_\delta).
\end{aligned} \tag{3.1}$$

We are able to identify and classify the set of fixed points of this effective dynamics. (Recall that the dynamics transits ballistically between these fixed points in $\delta_n^{-1}$ many steps.) We focus on the critical step-size regime with $c_\delta = 1$ where one sees from (3.1) that $u_2 \to 1$, which is where a random vector in $\mathbb{R}^n$ lies, and where the problem in the matrix case is most directly related to an eigenvalue problem (see Appendix D for the generic $c_\delta$ dependencies).

**Proposition 3.2.** *Eq. (3.1) has isolated fixed points classified as follows. Let $\lambda_c(k)$ be as in (D.4) and $m_\dagger(k, \lambda) \leq m_\star(k, \lambda)$ be as in (D.5) (if $k = 2$, $\lambda_c = 1$ and $m_\dagger = m_\star = \sqrt{\lambda - 1}$):*

1. *An unstable fixed point at $(0, 0)$ and a fixed point at $(0, 1)$; if $k = 2$, $(0, 1)$ is stable if $\lambda < \lambda_c(2)$ and unstable if $\lambda > \lambda_c(2)$; if $k > 2$ $(0, 1)$ is always stable.*

---

[2]One might also wish to rescale time like $\delta_n^{-\beta}$, where $\beta$ may depend on $t$; we leave this to future work.
[3]Note that one might also add additional penalty terms. The case of a ridge penalty is treated in Section D.
[4]The sub-critical regime of $\delta_n = o(1/n)$ is recovered by sending $c_\delta \to 0$ in the below.

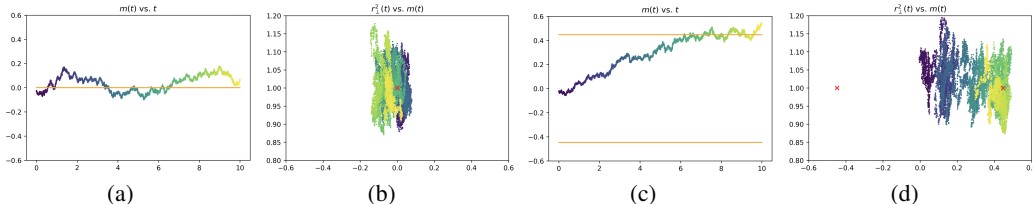

Figure 1: Matrix PCA summary statistics in dim. $n = 1500$ run for $10n$ steps at $\lambda = 0.8 < \lambda_c$ in (a)–(b) and $\lambda = 1.2 > \lambda_c$ in (c)–(d). Here, $\times$ and $-$ mark the stable fixed points of the systems. (a) and (c) demonstrate the stable and unstable OU processes that arise as diffusive limits of the $m$ variable, and (b) and (d) depict the trajectories in $(m, r_\perp^2)$ space.

2. *If $\lambda > \lambda_c(k)$: when $k = 2$, two stable fixed points at $(\pm m_\star(2), 1)$. When $k \geq 3$, two unstable fixed points at $(\pm m_\dagger(k), 1)$ and two stable fixed points at $(\pm m_\star(k), 1)$.*

*Remark* 1. The presence of *two* pairs of fixed points when $k \geq 3$ with non-zero correlation with $v$ may seem surprising—indeed it indicates that even some warm starts will fail to attain good correlation with the signal when $\lambda$ is finite. This is an interesting consequence of the corrector in (3.1) and if one tracked the $c_\delta$ dependence in the above, the fixed point $m_\dagger$ goes to zero as $c_\delta \to 0$ and this barrier to recovery from warm starts vanishes as one approaches sub-critical step-sizes.

Let us now consider a rescaling of $\mathbf{u}_n$ in a microscopic neighborhood of the saddle set $m = 0$. This captures the initial phase from a random start: if $\mu_n \sim \mathcal{N}(0, I_n/n)$, then $(\mathbf{u}_n)_* \mu_n \to \delta_{(0,1)}$ weakly. Now rescale and let $\tilde{\mathbf{u}}_n = (\sqrt{n}m, r_\perp^2)$. Evidently, $\tilde{\nu} = \lim_n (\tilde{\mathbf{u}}_n)_* \mu_n = \mathcal{N}(0, 1) \otimes \delta_1$.

**Proposition 3.3.** *Fix $k \geq 2$, $\lambda > 0$ and $\delta_n = 1/n$. Then $\tilde{\mathbf{u}}_n(t)$ converges as $n \to \infty$ to the solution of the following SDE initialized from $\tilde{\nu}$:*

$$d\tilde{u}_1 = 2\tilde{u}_1(2\lambda\mathbf{1}_{k=2} - k\tilde{u}_2^{k-1})dt + 2(k\tilde{u}_2^{k-1})^{1/2}dB_t$$
$$d\tilde{u}_2 = -4k\tilde{u}_2^{k-1}(\tilde{u}_2 - 1)dt\,. \tag{3.2}$$

We see that $\tilde{u}_2$ solves an autonomous ODE which converges exponentially to 1. When $k = 2$, the equation for $\tilde{u}_1$ then converges to $4(\lambda - 1)\tilde{u}_1 dt + 2\sqrt{2}dB_t$ for large $t$. This is an OU process which is stable when $\lambda < 1$ and unstable when $\lambda > 1$. By stitching together the prelimits of these OU processes at a sequence of scales interpolating between that of $\tilde{\mathbf{u}}_n$ and $\mathbf{u}_n$, one could in principle establish that for any $\lambda > 1$, SGD reaches the stable fixed points at $(\pm m_\star(2), 1)$ in $O(n \log n)$ steps (with precise asymptotics, etc.), while when $\lambda < 1$, the mean-reverting nature of the OU suggests it needs a much larger number of samples in order to correlate with the vector $v$. See Figure 1 for numerical verification of this intuition. When $k \geq 3$, the tensor PCA problem is known to be hard for SGD to solve without a polynomially diverging sample complexity or $\lambda$ [3]. Accordingly, when $\lambda$ is kept finite in $n$, the expression for $\tilde{u}_1$ in (3.2) is *always* a stable OU-type process. Interestingly, one can also capture the (diverging) signal-to-noise threshold for SGD to recover $v$ in tensor PCA by our methods. Indeed, for $k \geq 3$ if one considers $\lambda_n = \Lambda n^{(k-2)/2}$ (matching the predicted gradient-based algorithm threshold from [4]), $\tilde{\mathbf{u}}_n$ would instead converge to the solution of

$$d\tilde{u}_1 = 2\tilde{u}_1(k\Lambda - k\tilde{u}_2^{k-1})dt + 2(k\tilde{u}_2^{k-1})^{1/2}dB_t\,,$$
$$d\tilde{u}_2 = -4k\tilde{u}_2^{k-1}(\tilde{u}_2 - 1)dt\,,$$

which transitions between stable and unstable OU processes at $\Lambda_c(k) = 1$, as in the matrix case.

## 4 Two-layer networks for classifying a binary Gaussian mixture

As our second example, we consider the problem of supervised classification of a binary Gaussian mixture model (binary GMM) using a two-layer network. Our goal here is to demonstrate how our approach can be used to analyze the performance of SGD for multi-layer networks, and indeed we will find the calculations here to be relevant in Section 5 where we consider XOR-type GMM's.

Let us now formalize the problem. Suppose that we are given i.i.d. samples of the form $Y = (y, X)$, where $y$ is a $Ber(1/2)$ random variables and, conditionally on $y$, we have $X \sim \mathcal{N}((2y - 1)\mu, I/\lambda)$,

where $\mu \in \mathbb{R}^N$ is a fixed unit vector, $I$ is the identity on $\mathbb{R}^N$, and $\lambda > 0$ is the signal-to-noise ratio. Here, $y$ is the class label and $X$ is the data.

For the sake of concreteness, we consider classification via the following architecture (though our techniques generalize to other settings *mutatis mutandis*): The first layer has weights $(W_1, W_2) \in \mathbb{R}^N \times \mathbb{R}^N$ and ReLu activation, $g(x) = x \vee 0$; and the second layer has weights $v_1, v_2 \in \mathbb{R}$ and sigmoid activation, $\sigma(x) = 1/(1+e^{-x})$. Our parameter space is then $\mathcal{X}_n = \mathbb{R}^{2N+2}$ and we therefore take $n = 2N + 2$ when applying Theorem 2.2. As we are interested in supervised classification, we take the usual *binary cross-entropy loss* with $\ell^2$ regularization,

$$L\big((v_i, W_i)_{i \in \{1,2\}}; (y, X)\big) = -yv \cdot g(WX) + \log(1 + e^{v \cdot g(WX)}) + p(v, W), \qquad (4.1)$$

where $g$ is applied component wise and $p(v, W) := (\alpha/2)(||v||^2 + ||W||^2)$.

It can be shown (see Lemma E.1) that the law of the loss at a given point, $(v, W) \in \mathcal{X}_n$, depends only on the 7 summary statistics,

$$\mathbf{u}_n = (v_1, v_2, m_1, m_2, R_{11}^{\perp}, R_{12}^{\perp}, R_{22}^{\perp}), \qquad (4.2)$$

where $m_i = W_i \cdot \mu$ and $R_{ij}^{\perp} = W_i^{\perp} \cdot W_j^{\perp}$ with $W_i^{\perp} = W_i - m_i \mu$ denoting the part of $W_i$ orthogonal to $\mu$. For a point, $(v, W) \in \mathcal{X}_n$, let

$$
\begin{aligned}
\mathbf{A}_i^{\mu} &= \mathbb{E}[X \cdot \mu \mathbf{1}_{W_i \cdot X \geq 0}(\sigma(v \cdot g(WX)) - y)], \\
\mathbf{A}_{ij}^{\perp} &= \mathbb{E}[X \cdot W_j^{\perp} \mathbf{1}_{W_i \cdot X \geq 0}(\sigma(v \cdot g(WX)) - y)], \\
\mathbf{B}_{ij} &= \mathbb{E}[\mathbf{1}_{W_i \cdot X \geq 0} \mathbf{1}_{W_j \cdot X \geq 0}(\sigma(v \cdot g(WX)) - y)^2].
\end{aligned}
\qquad (4.3)
$$

By Lemma E.1, these are functions only of $\mathbf{u}_n$, and we denote them as such, e.g., $\mathbf{A}_i^{\mu} = \mathbf{A}_i^{\mu}(\mathbf{u}_n)$. The critical scaling for $\delta$ is then of order $\Theta(1/n)$ and we obtain the following effective dynamics.

**Proposition 4.1.** *Let $\mathbf{u}_n$ be as in (4.2) and fix any $\lambda > 0$ and $\delta_n = {}^{c_\delta}/N$. Then $\mathbf{u}_n(t)$ converges to the solution of the ODE system, $\dot{\mathbf{u}}_t = -\mathbf{f}(\mathbf{u}_t) + \mathbf{g}(\mathbf{u}_t)$, initialized from $\lim_{n \to \infty}(\mathbf{u}_n)_* \mu_n$, with:*

$$
\begin{aligned}
f_{v_i} &= m_i \mathbf{A}_i^{\mu}(\mathbf{u}) + \mathbf{A}_{ii}^{\perp}(\mathbf{u}) + \alpha v_i, \\
f_{m_i} &= v_i \mathbf{A}_i^{\mu}(\mathbf{u}) + \alpha m_i, \\
f_{R_{ij}^{\perp}} &= v_i \mathbf{A}_{ij}^{\perp}(\mathbf{u}) + v_j \mathbf{A}_{ji}^{\perp}(\mathbf{u}) + 2\alpha R_{ij}^{\perp},
\end{aligned}
$$

*and correctors $g_{v_i} = g_{m_i} = 0$, $g_{R_{ij}^{\perp}} = c_\delta \frac{v_i v_j}{\lambda} \mathbf{B}_{ij}$ for $i, j = 1, 2$.*

Due to the Gaussian integrals defining $\mathbf{f}, \mathbf{g}$, it is difficult to analyze the ODE system defined by Proposition 4.1, let alone any rescaled effective dynamics. For ease of analysis, we next send $\lambda \to \infty$ corresponding to a small noise regime for the Gaussian mixture. We emphasize that this limit is taken after $n \to \infty$ and therefore is still approximately on the critical scale of $\lambda = \Theta(1)$ at which there is a transition in the existence of any fixed point which is a good classifier. In particular, if $\lambda = \lambda_n$ is any diverging sequence, then the limiting effective dynamics would exactly match that attained by now sending $\lambda \to \infty$. In Figure 2, we demonstrate numerically that the following predicted fixed points from the $\lambda \to \infty$ limit match those arising at finite large $n$ and $\lambda > 0$.[5]

**Proposition 4.2.** *The $\lambda \to \infty$ limit of the ODE system of Proposition 4.1 is given by*

$$
\dot{m}_i = \begin{cases} \frac{v_i}{2}\sigma(-v \cdot m) - \alpha m_i & m_1 m_2 > 0 \\ \frac{v_i}{2}\sigma(-v_i m_i) - \alpha m_i & else \end{cases},
$$

$$
\dot{v}_i = \begin{cases} \frac{m_i}{2}\sigma(-v \cdot m) - \alpha v_i & m_1 m_2 > 0 \\ \frac{m_i}{2}\sigma(-v_i m_i) - \alpha v_i & else \end{cases},
$$

*and $\dot{R}_{ij}^{\perp} = -2\alpha R_{ij}^{\perp}$. The fixed points of this system are classified as follows. All fixed points have $R_{ij}^{\perp} = 0$ and $m_i = v_i$ for $i, j = \{1, 2\}$. In $(v_1, v_2)$, the coordinates are classified by*

    *1. A fixed point at $(v_1, v_2) = (0, 0)$ that is stable if $\alpha > {}^1/4$.*

---

[5]For large $\lambda$, this is indeed a quantitative approximation as $\mathbf{f}, \mathbf{g}$ exhibit locally Lipschitz dependence on $\lambda^{-1}$, so the corresponding dynamics converges as $\lambda \to \infty$ by classical well-posedness results (see, e.g., [48])

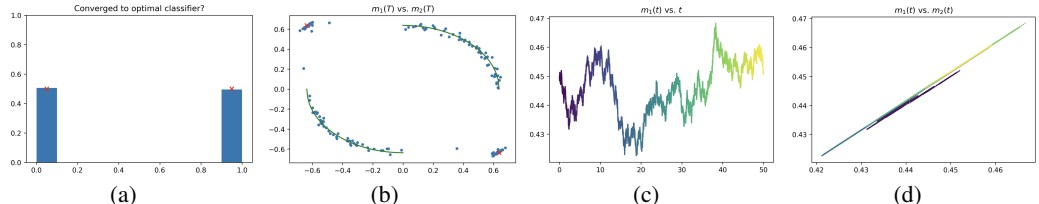

Figure 2: Binary GMM in dim. $N = 250$ with $\lambda = 100$ and $\alpha = 0.1$. (a) fraction of runs converging to the optimal classifier ($\times$ marking the predicted $1/2$ fraction), and (b) endpoints of $(m_1, m_2)$ in 200 runs ($\times$ denoting the $\lambda = \infty$ predicted stable fixed points and $-$ the unstable rings). (c)–(d) diffusive limits, first for $m_1$, and then for the pair $m_1, m_2$ where the diffusion can be seen to be of rank 1.

2. If $\alpha < 1/4$, two unstable sets of fixed points at the quarter-circles given by $(v_1, v_2)$ having $v_1 v_2 > 0$ such that $v_1^2 + v_2^2 = C_\alpha$ for $C_\alpha := \log(1 - 2\alpha) - \log(2\alpha)$.

3. If $\alpha < 1/4$, two stable fixed points at $(v_1, v_2)$ equals $(\sqrt{C_\alpha}, -\sqrt{C_\alpha})$ and $(-\sqrt{C_\alpha}, \sqrt{C_\alpha})$.

*If $\mu_n$ is e.g., given by $(v_1, v_2) \sim \mathcal{N}(0, I_2)$ and $W_1, W_2 \sim \mathcal{N}(0, I_N/(\lambda N))$ then $\nu := \lim(\mathbf{u}_n)_* \mu_n$ is $\mathcal{N}(0, I_2)$ in the $v_1, v_2$ coordinates, and is in the basin of attraction of the quarter-circles of item (2) with probability $1/2$ and the basin of attraction of the stable fixed points of (3) with probability $1/2$.*

Let us pause to interpret this result. The stable fixed points when $\alpha < 1/4$ are the optimal classifiers, whereas the unstable set of fixed points given by item (2) misclassify half of the data. Therefore, the above indicates that when solving the above task with randomly initialized weights, one of the following two scenarios occur, each with probability $1/2$ (w.r.t. the initialization): the algorithm will converge to the optimal classifier in linear time or it will appear to have converged to a macroscopically sub-optimal classifier on the same timescale, see Figure 2(a)–(b).

It is then natural to ask about the behaviour of the SGD in the latter regime, after it converges to the sub-optimal classifiers which lie on the aforementioned quarter-circles. Proposition 4.2 rigorously justified the exchange of $n \to \infty$ and $\lambda \to \infty$ limits in the ballistic phase. In the diffusive phase, one could in principle find the quarter circle of fixed points of the ODE in Proposition 4.1 and consider rescaled observables $\tilde{v}_i, \tilde{m}_i$ corresponding to blowing up $v_i, m_i$ in diffusive $O(n^{-1/2})$ neighborhoods about them to get SDE limits from Theorem 2.2. In order to have explicit formulae, in what follows, we consider the diffusive limits obtained when taking $\lambda = \infty$, for which we know the precise locations of these fixed points from Proposition 4.2. This also captures the limit obtained by taking any $\lambda_n$ diverging faster than $O(n^{1/2})$; the numerics of Figure 2(c)–(d) demonstrate its qualitative consistency with the behavior in microscopic neighborhoods of fixed points even at $\lambda$ finite.

**Proposition 4.3.** *Let $\delta_n = 1/N$, $(a_1, a_2) \in \mathbb{R}_+^2$ be such that $a_1^2 + a_2^2 = C_\alpha$ and let $\tilde{v}_i = \sqrt{N}(v_i - a_i)$ and $\tilde{m}_i = \sqrt{N}(m_i - a_i)$. When $\lambda = \infty$, the SDE system obtained by applying Theorem 2.2 to $\tilde{\mathbf{u}}_n$ is*

$$d\tilde{v}_i = \alpha(\tilde{m}_i - \tilde{v}_i) + a_i(\alpha - 2\alpha^2) \sum a_k(\tilde{v}_k + \tilde{m}_k) + \tilde{\Sigma}^{1/2} d\mathbf{B}_t \cdot e_{v_i},$$

$$d\tilde{m}_i = \alpha(\tilde{v}_i - \tilde{m}_i) + a_i(\alpha - 2\alpha^2) \sum a_k(\tilde{v}_k + \tilde{m}_k) + \tilde{\Sigma}^{1/2} d\mathbf{B}_t \cdot e_{m_i},$$

$$\dot{R}_{i\perp} = -\alpha R_{i\perp} \qquad\qquad \dot{R}_{12}^\perp = -2\alpha R_{12}^\perp,$$

*where $\tilde{\Sigma}$ is a constant matrix whose only non-zero entries are $\tilde{\Sigma}_{\tilde{v}_i \tilde{v}_j} = \tilde{\Sigma}_{\tilde{m}_i \tilde{m}_j} = \tilde{\Sigma}_{\tilde{v}_i \tilde{m}_j} = \alpha^2 a_i a_j$.*

Notice that this diffusion matrix is rank 1, so this diffusion is non-trivial but degenerate even in the rescaled coordinates $(\tilde{v}_i, \tilde{m}_i)$. Moreover, the entries of $\tilde{\Sigma}$ vanish on the axes $a_1 = 0$ or $a_2 = 0$. In particular, crossing from the unstable quarter ring into the quadrants $v_1 v_2 < 0$ where the stable fixed points lie is *impossible* in the noiseless setting, and happens on a much larger timescale at finite $\lambda$.

## 5 Two-layer networks for the XOR Gaussian mixture

We end by applying our methods to supervised learning for an XOR-type Gaussian mixture model in $\mathbb{R}^N$ with a two-layer network. The ballistic phase of this problem was recently studied in [37].

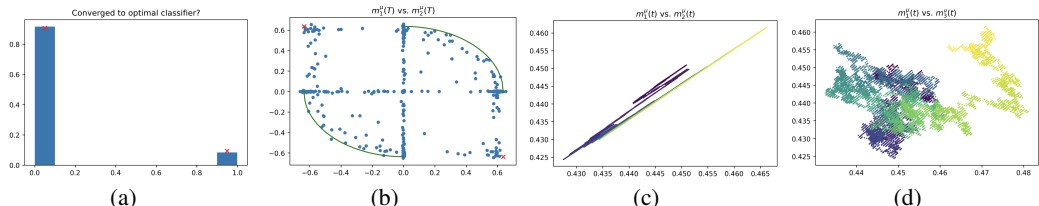

$$\text{(a)} \qquad\qquad \text{(b)} \qquad\qquad \text{(c)} \qquad\qquad \text{(d)}$$

Figure 3: XOR GMM in dim. $N = 250$ with $\lambda = 1000$ and $\alpha = 0.1$. (a) Fraction of runs converging to the optimal classifier ($\times$ marking the predicted $^{29}/_{32}$ and $^3/_{32}$) and (b) endpoints of $(m_1^\mu, m_2^\mu)$ in 200 runs ($\times$ denoting the $\lambda = \infty$ predicted stable fixed points, $-$ the unstable rings). (c)–(d) display the rank-2 diffusive limits in the regime of Proposition 5.1 in $(m_1^\mu, m_2^\mu)$ and $(m_1^\mu, m_3^\nu)$ coordinates.

Suppose that we are given i.i.d. samples of the form $Y = (y, X)$, where $y$ is $Ber(1/2)$ as before and $X$ has the following distribution: if $y = 1$ then $X$ is a $^1/_2$-$^1/_2$ mixture of $\mathcal{N}(\mu, I/\lambda)$ and $\mathcal{N}(-\mu, I/\lambda)$ and if $y = 0$ it is a $^1/_2$-$^1/_2$ mixture of $\mathcal{N}(\nu, I/\lambda)$ and $\mathcal{N}(-\nu, I/\lambda)$, where $\lambda > 0$, and $\mu, \nu$ are orthogonal unit vectors. Here, $y$ is the class membership label and $X$ is the data.

Consider the corresponding classification problem using a two-layer neural network, $\hat{y}(X) = \sigma(v \cdot g(WX))$, where $\sigma$ and $g$ are as in Section 4. We take $W$ to be a $4 \times N$ matrix and $v$ to be a 4-vector, and consider the binary cross-entropy loss as before. It is shown in Lemma F.1 that the law of the loss at a point $(v, W)$ depends only on the following 22 variables: for $1 \le i \le j \le 4$,

$$v_i, \qquad m_i^\mu = W_i \cdot \mu, \qquad m_i^\nu = W_i \cdot \nu, \qquad R_{ij}^\perp = W_i^\perp \cdot W_j^\perp \qquad (5.1)$$

where $W_i^\perp = W_i - m_i^\mu \mu - m_i^\nu \nu$ is the part perpendicular to $\mu, \nu$. With the choice of $\mathbf{u}_n$ given by these variables, for any fixed $\lambda > 0$, the localizability criterion of Definition 2.1 can be verified to hold as long as $\delta_n = O(1/n)$. In particular, we can apply Theorem 2.2 to obtain limits in both the ballistic and diffusive phases. For the precise equations in the ballistic phase, see Proposition F.2.

The fixed points of the ballistic dynamics, again in the limit $\lambda \to \infty$ after $n \to \infty$, are classified as follows (see Proposition F.3). If $\alpha > ^1/_8$, then the only fixed point is at $\mathbf{u}_n = \mathbf{0}$. If $0 < \alpha < ^1/_8$, then let $(I_0, I_\mu^+, I_\mu^-, I_\nu^+, I_\nu^-)$ be any disjoint (possibly empty) subsets whose union is $\{1, \dots, 4\}$. Each such partition corresponds to a connected component of fixed points. Corresponding to a such tuple, the connected component of fixed points has $R_{ij}^\perp = 0$ for all $i, j$, and

1. $m_i^\mu = m_i^\nu = v_i = 0$ for $i \in I_0$,
2. $m_i^\mu = v_i > 0$ such that $\sum_{i \in I_\mu^+} v_i^2 = -\text{logit}(4\alpha)$ and $m_i^\nu = 0$ for all $i \in I_\mu^+$,
3. $-m_i^\mu = v_i > 0$ such that $\sum_{i \in I_\mu^-} v_i^2 = -\text{logit}(4\alpha)$ and $m_i^\nu = 0$ for all $i \in I_\mu^-$,
4. $m_i^\nu = v_i < 0$ such that $\sum_{i \in I_\nu^+} v_i^2 = -\text{logit}(4\alpha)$ and $m_i^\mu = 0$ for all $i \in I_\nu^+$,
5. $-m_i^\nu = v_i < 0$ such that $\sum_{i \in I_\nu^-} v_i^2 = -\text{logit}(4\alpha)$ and $m_i^\mu = 0$ for all $i \in I_\nu^-$.

There are 39 connected components of fixed points, 4! of which are stable, one for each permutation where $I_\mu^+, I_\mu^-, I_\nu^+, I_\nu^-$ are all singletons. For the proof of this limit, see Proposition F.3.

We can also compute the probability that the effective dynamics in the ballistic phase converges to a stable fixed point (as opposed to an unstable one). From a Gaussian initialization $\mu_n$ where $v_i \sim \mathcal{N}(0, 1)$ and $W_i \sim \mathcal{N}(0, I_N/N)$ independently, this probability will converge to $^3/_{32}$.

As an example of the diffusions that can arise in the rescaled effective dynamics at the unstable fixed points, let us consider the unstable fixed points in which $v$ has the correct signature (two positive, two negative) but for each of those we are at a corresponding quarter-ring. Here, the dynamics effectively becomes a pair of 2 two-layer GMM's on quarter-rings (as in Section 4), that are anti-correlated. More precisely, let $(a_{1,\mu}, a_{2,\mu})$ be such that $a_{1,\mu}^2 + a_{2,\mu}^2 = C_\alpha$ and $(a_{3,\nu}, a_{4,\nu})$ such that $a_{3,\nu}^2 + a_{4,\nu}^2 = C_\alpha$, for $C_\alpha = -\text{logit}(4\alpha)$. Take as fixed points about which we expand to be $v_i = m_i^\mu = a_{i,\mu} > 0$ and $v_i = m_i^\nu = a_{i,\nu} < 0$ for $i = 3, 4$. Namely, we let

$$\tilde{v}_i = \begin{cases} \sqrt{N}(v_i - a_{i,\mu}) & i = 1, 2 \\ \sqrt{N}(v_i - a_{i,\nu}) & i = 3, 4 \end{cases},$$

and let

$$\tilde{m}_i^\mu = \sqrt{N}(m_i^\mu - a_{i,\mu}) \quad i = 1, 2$$
$$\tilde{m}_i^\nu = \sqrt{N}(m_i^\nu - a_{i,\nu}) \quad i = 3, 4 \,.$$

(We set $\tilde{m}_i^\nu = 0$ for $i = 1, 2$ and $\tilde{m}_i^\mu = 0$ for $i = 3, 4$ in $\tilde{\mathbf{u}}_n$ effectively removing those variables.)

**Proposition 5.1.** *Let $\delta_n = {}^1/_N$ and let $\tilde{\mathbf{u}}_n = (\tilde{v}_i, \tilde{m}_i^\mu, \tilde{m}_i^\nu, R_{ij}^\perp)$. When $\lambda = \infty$, Theorem 2.2 can be applied and $\tilde{\mathbf{u}}_n(t)$ converges to the solution of the SDE $d\tilde{\mathbf{u}}(t) = -\tilde{\mathbf{h}}(\tilde{\mathbf{u}})dt + \sqrt{\Sigma(\tilde{\mathbf{u}})}d\mathbf{B}_t$ where*

$$\tilde{h}_{\tilde{v}_i} = \begin{cases} \alpha(\tilde{v}_i - \tilde{m}_i^\mu) - a_{i,\mu}(\alpha - 4\alpha^2) \sum_{k=1,2} a_{k,\mu}(\tilde{v}_k + \tilde{m}_k^\mu) & i = 1, 2 \\ \alpha(\tilde{v}_i - \tilde{m}_i^\nu) - a_{i,\nu}(\alpha - 4\alpha^2) \sum_{k=3,4} a_{k,\nu}(\tilde{v}_k + \tilde{m}_k^\nu) & i = 3, 4 \end{cases} ,$$

*$\tilde{h}_{\tilde{m}_i^\mu}$ (resp., $\tilde{h}_{\tilde{m}_i^\nu}$) is like $h_{\tilde{v}_i}$ for $i = 1, 2$ (resp., $i = 3, 4$) with $\tilde{v}_i$ and $\tilde{m}_i^\mu$ (resp., $\tilde{m}_i^\nu$) swapped, $\tilde{h}_{R_{ij}^\perp} = 2\alpha R_{ij}^\perp$, and $\tilde{\Sigma}$ is the constant rank-2 matrix whose non-zero entries are*

$$\tilde{\Sigma}_{\tilde{v}_i \tilde{v}_j} = \tilde{\Sigma}_{\tilde{m}_i^\mu \tilde{m}_j^\mu} = \tilde{\Sigma}_{\tilde{v}_i \tilde{m}_j^\mu} = 3\alpha^2 a_{i,\mu} a_{j,\mu} \quad \text{if } i, j \in \{1, 2\} \,,$$
$$\tilde{\Sigma}_{\tilde{v}_i \tilde{v}_j} = \tilde{\Sigma}_{\tilde{m}_i^\nu \tilde{m}_j^\nu} = \tilde{\Sigma}_{\tilde{v}_i \tilde{m}_j^\nu} = 3\alpha^2 a_{i,\nu} a_{j,\nu} \quad \text{if } i, j \in \{3, 4\} \,,$$
$$\tilde{\Sigma}_{\tilde{v}_i \tilde{v}_j} = \tilde{\Sigma}_{\tilde{m}_i^\mu \tilde{m}_j^\nu} = \tilde{\Sigma}_{m_i^\mu, v_j} = \tilde{\Sigma}_{\tilde{v}_i \tilde{m}_j^\nu} = -\alpha^2 a_{i,\mu} a_{j,\nu} \quad \text{if } i \in \{1, 2\}, j \in \{3, 4\} \,.$$

## Acknowledgements

The authors thank the anonymous referees for their careful readings of the manuscript and helpful comments. R.G. thanks the Miller Institute for Basic Research in Science for its support. A.J. acknowledges the support of the Natural Sciences and Engineering Research Council of Canada (NSERC). Cette recherche a été financée par le Conseil de recherches en sciences naturelles et en génie du Canada (CRSNG), [RGPIN-2020-04597, DGECR-2020-00199].

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
