# A    Notation

For the reader's reference, we collect the notation that recurs throughout this paper.

The data space $\mathcal{Y}_n$ is a subset of $\mathbb{R}^{d_n}$ and the parameter space $\mathcal{X}_n$ is a subset of $\mathbb{R}^{p_n}$. We let $\langle \cdot, \cdot \rangle$ be the Euclidean inner product of its inputs, with the space understood contextually.

$Y_\ell$ denotes the $\ell$'th data point drawn from the data distribution $P_n \in \mathcal{M}(\mathcal{Y}_n)$ where $\mathcal{M}(A)$ is the set of all probability measures on $A$. $X_\ell$ denotes the stochastic gradient descent at step $\ell$, generated with loss function $L : \mathcal{X}_n \times \mathcal{Y}_n \to \mathbb{R}$ and step-size $\delta_n$. We consider $(X_\ell)$ initialized from $X_0 \sim \mu_n$ for $\mu_n \in \mathcal{M}(\mathcal{X}_n)$.

For a given loss function $L$, we let $\Phi(x) = \mathbb{E}L(x, Y)$ denote the population loss, and let $H(x)$ be the random variable $L(x, Y) - \Phi(x)$. We can then define, at each point $x \in \mathbb{R}^{p_n}$, the covariance matrix

$$V(x) = \mathbb{E}[\nabla H(x) \otimes \nabla H(x)],$$

where $\otimes$ denotes the tensor product, and the first and second order differential operators

$$\mathcal{A}_n = \langle \nabla H, \nabla \rangle = \sum_i \partial_i \Phi(x) \partial_i \qquad \text{and} \qquad \mathcal{L}_N = \frac{1}{2} \langle V(x), \nabla^2 \rangle = \frac{1}{2} \sum_{i,j} V_{ij}(x) \partial_i \partial_j .$$

In our paper, $\mathbf{u}_n(x) = (u_1(x), ..., u_k(x))$ denotes the summary statistics, with $\mathbf{u}_n \in C^1(\mathbb{R}^{p_n}; \mathbb{R}^k)$ meaning it is a continuously differentiable function from $\mathbb{R}^{p_n}$ to $\mathbb{R}^k$. We let $J_n$ be the corresponding Jacobian.

Given the above, when the limiting dynamics exist per Theorem 2.2, the limiting dynamics is denoted $\mathbf{u}_t$ (with the subscript $n$ dropped), $\mathbf{h}$ is the limiting effective drift defined as in (2.1), and $\Sigma(u)$ denotes the limiting volatility as in (2.2). $\mathbf{B}_t$ denotes standard brownian motion in $\mathbb{R}^k$. The initialization for the limiting dynamics is given by $\nu = \lim_{n \to \infty} (\mathbf{u}_n)_* \mu_n$.

# B    Remarks and intuition on localizability conditions

Let us include a brief discussion of the various scalings appearing in Definition 2.1.

*Remark* 2. The kinds of summary statistics that we most frequently have in mind for application are (1) linear functions of the parameter space $\mathcal{X}_n$, for instance the correlation with a unit vector, or some ground truth; (2) radial statistics, like the $\ell^2$-norm of the parameters, or some subset of the parameters; and (3) rescaled versions (usually blown up by $\delta_n^{-1/2}$ of these near their fixed points. Regarding the item (1) in Definition 2.1, for linear functions, it trivially holds; for radial statistics, the Hessian is a block identity matrix, so item (1) holds as long as $\delta_n$ is $O(1)$; therefore item (1) is most restrictive for rescalings of non-linear statistics, e.g., $u(x) = \delta_n^{-\alpha}(\|x\|^2 - 1)$ where it prevents consideration of this statistic with $\alpha > 1/2$.

Turning to item (2) of Definition 2.1, we comment that the regularity assumptions made on $L$ here are less restrictive than uniform Lipchitz or smoothness assumptions common to the literature. In particular, we do not assume the population loss is Lipschitz everywhere, as we may have that $\bigcup_K \mathbf{u}_n^{-1}(E_K)$ does not cover $\mathcal{X}_n$, nor does it imply uniform smoothness of $H$ (and in turn $L$) as we will be taking $\delta_n \to 0$ with $n$.

Let us lastly motivate the scalings appearing in item (3), which ensure there is some independence between $H$ and the values of $\nabla u$ and $\nabla^2 u$ at $x$. As a testbed, suppose that $\nabla H(x)$ is a random vector with i.i.d. entries all of order 1. Then, if $u$ is a rescaled linear statistic, e.g., $\delta_n^{-1/2} \langle x, e_1 \rangle$ then the first bound of item (3) is saturated, and the second of course is trivial due to the linearity of $u$. The second bound is saturated by taking a rescaling of a radial statistic, e.g., $\delta_n^{-1/2} \|x\|^2$, again assuming for maximal simplicity that $\nabla H$ is an i.i.d. random vector with order one entries. In fact, the second part of item (3) could be dropped at the expense of more complicated diffusion coefficients in limiting SDE's: see Remark 3.

*Remark* 3. While we discussed above the reasons for which the various scalings of Definition 2.1 were selected, it is interesting to ask what changes in Theorem 2.2 should certain of the assumptions of Definition 2.1 be violated. Most of the assumed bounds in the definition of localizability are used to establish tightness and ensure higher order terms in Taylor expansions vanish in the $n \to \infty$

limit. One assumption which could in principle be relaxed is the second assumption in item (3) of Definition 2.1; indeed if that assumption is dropped, Theorem 2.2 would still essentially apply, except the limiting diffusion matrix would be the $n \to \infty$ limit (assuming it exists) of

$$\delta J V J^T + \delta J \mathbb{E}[\langle \nabla H, J \rangle \otimes \langle \nabla^2 \mathbf{u}, \nabla H \otimes \nabla H - V \rangle] + \delta J \mathbb{E}[\langle \nabla^2 \mathbf{u}, \nabla H \otimes \nabla H - V \rangle \otimes \langle \nabla H, J \rangle]$$
$$+ \delta^2 \mathbb{E}[\langle \nabla^2 \mathbf{u}, \nabla H \otimes \nabla H - V \rangle^{\otimes 2}],$$

as opposed to simply the limit of $\delta J V J^T$.

## C   Proof of Theorem 2.2

In this section, we prove our main convergence result, namely Theorem 2.2. The proof of this can be thought as a version of the classical *martingale problem* [46] for summary statistics of stochastic gradient descent in the high-dimensional $n \to \infty$ limit.

For ease of notation, in the following we say that $f \lesssim g$ if there is some constant $C > 0$ such that $f \leq Cg$ and that $f \lesssim_a g$ if there is some constant $C(a) > 0$ depending only on $a$ such that $f \leq C(a)g$. Furthermore, for readability, we will often suppress the dependence on $n$ in subscripts, when it is clear from context.

**Proof of Theorem 2.2.**  Our aim is to establish $\mathbf{u}_n \to \mathbf{u}$ weakly as random variables on $C([0, \infty))$ where $\mathbf{u}$ solves (2.3). It is equivalent to show the same on $C([0, T])$ for every $T > 0$.

Let $\tau_K^n$ denote the exit time for the interpolated process $\mathbf{u}_n(t)$ from $E_K^n$. Define its pre-image $E_{K,n}^* := \mathbf{u}_n^{-1}(E_K^n)$ and let $L_{K,n}^\infty = L^\infty(E_{K,n}^*)$. For any function $f$, we use the shorthand $f_\ell$ to denote $f(X_\ell)$. By Taylor's theorem, we have that for any $C^3$ function $f$ and any $\ell \leq \tau_K^n/\delta$,

$$\begin{aligned} f_\ell &= f(X_{\ell-1} - \delta \nabla \Phi_{\ell-1} - \delta \nabla H_{\ell-1}^\ell) \\ &= f_{\ell-1} - \delta[A_\ell^f - A_{\ell-1}^f] - \delta[M_\ell^f - M_{\ell-1}^f] + O(\delta^3 \|\nabla^3 f\|_{L_{K,n}^\infty} \cdot \|\nabla L\|_{L_{K,n}^\infty}^3), \end{aligned} \quad \text{(C.1)}$$

where $A_\ell^f$ and $M_\ell^f$ are defined by their increments as follows:

$$A_\ell^f - A_{\ell-1}^f = \langle \nabla \Phi, \nabla f \rangle_{\ell-1} - \delta \left( \mathcal{L}_n f_{\ell-1} + \tfrac{1}{2} \langle \nabla \Phi \otimes \nabla \Phi, \nabla^2 f \rangle_{\ell-1} \right),$$
$$M_\ell^f - M_{\ell-1}^f = \langle \nabla H^\ell, \nabla f \rangle_{\ell-1} + \delta(\mathcal{E}_\ell^f - \mathcal{E}_{\ell-1}^f),$$
$$\mathcal{E}_\ell^f - \mathcal{E}_{\ell-1}^f = -\nabla^2 f(\nabla \Phi, \nabla H^\ell)_{\ell-1} - \tfrac{1}{2} \langle \nabla^2 f, \nabla H^\ell \otimes \nabla H^\ell - V \rangle_{\ell-1},$$

for $\mathcal{L}_n = \frac{1}{2} \sum_{i,j} V_{ij} \partial_i \partial_j$ and $V = \mathbb{E}[\nabla H \otimes \nabla H]$. Observe that $A_\ell^f$ is pre-visible and $M_\ell^f$ is a martingale. We bound these for $f = u_j$ among $\mathbf{u}_n = (u_1, ..., u_k)$.

After recalling Definition 2.1, we see that since $\mathbf{u}_n$ are $\delta_n$-localizable, the error term in (C.1) has

$$\delta^3 \sup_{x \in E_{K,n}^*} \mathbb{E}[\|\nabla^3 u_j\| \cdot \|\nabla L\|^3] \lesssim \delta^3 \|\nabla^3 u_j\|_{L_{K,n}^\infty} \left( \|\nabla \Phi\|_{L_{K,n}^\infty}^3 + \sup_{E_{K,n}^*} \mathbb{E}\|\nabla H\|^3 \right) \lesssim_K \delta^{3/2}.$$

Since $\delta_n$ goes to infinity as $n \to \infty$, we may thus write $u_j(X_\ell)$ as

$$u_j(X_\ell) = u_j(0) - \delta \sum_{\ell' \leq \ell} \left( A_{\ell'}^{u_j} - A_{\ell'-1}^{u_j} \right) - \delta \sum_{\ell' \leq \ell} \left( M_{\ell'}^{u_j} - M_{\ell'-1}^{u_j} \right) + o(1),$$

where the last term is $o(1)$ in $L^1$ uniformly for $\ell \leq \tau_K/\delta$.

Now let us define for $s \in [0, T]$,

$$a_j'(s) = A_{[s/\delta]}^{u_j} - A_{[s/\delta]-1}^{u_j}$$
$$b_j'(s) = M_{[s/\delta]}^{u_j} - M_{[s/\delta]-1}^{u_j}$$

If we let $a_j(s) = \int_0^s a_j'(s') ds' = a_j(\delta[s/\delta]) + (s - \delta[s/\delta])(A_{[s/\delta]}^{u_j} - A_{[s/\delta]-1}^{u_j})$ and $b_j(s) = \int_0^s b_j'(s') ds'$, then recalling that $\mathbf{u}_n(s)$ is the linear interpolation of $(u_j([s/\delta]))_j$, we may write

$$\mathbf{u}_n(s) = \mathbf{u}_n(0) + \mathbf{a}_n(s) + \mathbf{b}_n(s) + o(1).$$

where $\mathbf{a}_n(s) = (a_j(s))_j$ and $\mathbf{b}_n(s) = (b_j(s))_j$.

We now prove that the sequence $(\mathbf{u}_n(s \wedge \tau_K^n))$ is tight in $C([0, T])$ with limit points which are $\alpha$-Holder for each $K$. To this end, let us define $\mathbf{v}_n(s) = \mathbf{a}_n(s) + \mathbf{b}_n(s) + \mathbf{u}_n(0)$. As the $o(1)$ error above is uniform in $t$, we have that

$$\sup_{0 \le s \le \tau_K^n \delta} ||\mathbf{u}_n(s) - \mathbf{v}_n(s)|| \to 0, \qquad \text{in } L^1.$$

Thus it suffices to show the claimed tightness and Holder properties of limit points for $\mathbf{v}_n$ instead of $\mathbf{u}_n$. We aim to show that for all $0 \le s, t \le T$,

$$\mathbb{E}||\mathbf{v}_n(s \wedge \tau_K) - \mathbf{v}_n(t \wedge \tau_K)||^4 \lesssim_{K,T} (t - s)^2, \tag{C.2}$$

from which we will get that the sequence $\mathbf{v}_n(s \wedge \tau_K)$ is uniformly $1/4$-Hölder by Kolmogorov's continuity theorem.

Evidently, for all $s, t$ we have

$$||\mathbf{v}_n(s) - \mathbf{v}_n(t)|| \le ||\mathbf{a}_n(s) - \mathbf{a}_n(t)|| + ||\mathbf{b}_n(s) - \mathbf{b}_n(t)||.$$

We control these terms in turn. We will do this coordinate wise and, for readability, fix some $j \le k$ and let $u = u_j$, $a = a_j$, $b = b_j$ etc.

For the pre-visible term, we have

$$\mathbb{E}|a(s \wedge \tau_K) - a(t \wedge \tau_K)|^4 \lesssim$$
$$\mathbb{E}\Big|\delta \sum_k \big(\langle \nabla\Phi, \nabla u \rangle_k - (\delta\mathcal{L}_n u)_k\big)\Big|^4 + \mathbb{E}\Big|\delta^2 \sum_k \langle \nabla\Phi \otimes \nabla\Phi, \nabla^2 u \rangle_k\Big|^4, \tag{C.3}$$

where these sums are over steps $k$ ranging from $[s/\delta] \wedge \tau_K/\delta$ to $[t/\delta] \wedge \tau_K/\delta$.

Let $\mathbf{h} = (h_j)_{j \le k}$ be as in (2.1). Then by (2.1), we have

$$\big|\langle \nabla\Phi(x), \nabla u(x) \rangle - (\delta\mathcal{L}_n u)(x)\big| \le h_j(\mathbf{u}_n(x)) + o(1),$$

uniformly over all $x \in E_{K,n}^*$. Therefore, the first term in (C.3) satisfies

$$\mathbb{E}\Big|\delta \sum_k \big(\langle \nabla\Phi, \nabla u \rangle_k - (\delta\mathcal{L}_n u)_k\big)\Big|^4 \lesssim \mathbb{E}\Big|\delta \sum h_j(\mathbf{u}_n)_k\Big|^4 + o((t-s)^4)$$

$$\le (t-s)^4 \left(||h_j||_{L^\infty(E_K^n)}^4 + o(1)\right)$$

$$\lesssim (t-s)^4$$

by continuity of $h_j$. For the second term in (C.3),

$$\mathbb{E}\Big|\delta^2 \sum \langle \nabla\Phi \otimes \nabla\Phi, \nabla^2 u \rangle_\ell\Big|^4 \le \delta^8 \Big(|((t-s)/\delta)| \sup_{x \in E_{K,n}^*} ||\nabla\Phi(x)||^2 \sup_{x \in E_{K,n}^*} ||\nabla^2 u(x)||_{op}\Big)^4$$

$$\lesssim_K \delta^2 (t-s)^4$$

where in the last inequality, we have used the definition of $\delta_n$-localizability. (Applying this bound for $s = 0, t = T$, the last term in $a$ is vanishing in the limit for each $K$ whenever $\delta_n = o(1)$.) Combining the above bounds yields

$$\mathbb{E}|a(s \wedge \tau_K) - a(t \wedge \tau_K)|^4 \lesssim_K (t-s)^4.$$

For the martingale term, notice that by independence,

$$\mathbb{E}|b(s \wedge \tau_K) - b(t \wedge \tau_K)|^4 = \mathbb{E}\left[\Big(\delta \sum(M_\ell^u - M_{\ell-1}^u)\Big)^4\right] = \mathbb{E}\left[\Big(\delta^2 \sum(M_\ell^u - M_{\ell-1}^u)^2\Big)^2\right],$$

where the sum again runs over steps $\ell$ ranging from $[s/\delta] \wedge \tau_K$ to $[t/\delta] \wedge \tau_K$. Repeatedly using the inequality $(x + y + z)^2 \lesssim x^2 + y^2 + z^2$, it suffices to bound the above quantity for each of the three terms defining the martingale difference $M_\ell^u - M_{\ell-1}^u$ respectively.

For the first term in that martingale difference, observe that

$$\mathbb{E}\Big[\Big(\delta^2 \sum_\ell \langle \nabla H^\ell, \nabla u \rangle_{\ell-1}^2\Big)^2\Big] = \delta^4 \sum_{\ell,\ell'} \mathbb{E}\Big[\langle \nabla H^\ell, \nabla u \rangle_{\ell-1}^2 \langle \nabla H^{\ell'}, \nabla u \rangle_{\ell'-1}^2\Big]$$

$$\le \Big(\delta \sum_\ell \big(\delta^2 \mathbb{E}\langle \nabla H^\ell, \nabla u \rangle_{\ell-1}^4\big)^{1/2}\Big)^2$$

$$\lesssim_K (t-s)^2, \tag{C.4}$$

where in the middle line we used Cauchy-Schwarz and in the last we used $\delta_n$-localizability.

For the second term in the martingale difference,

$$\mathbb{E}\Big[\Big(\delta^4 \sum_\ell \big(\nabla^2 u(\nabla\Phi, \nabla H^\ell)_{\ell-1}\big)^2\Big)^2\Big] \le \delta^6(t-s)^2 \sup_{x\in E_{K,n}^*} ||\nabla^2 u(x)||^4 \cdot ||\nabla\Phi(x)||^4 \cdot \mathbb{E}||\nabla H(x)||^4$$

$$\lesssim_K \delta^2(t-s)^2, \tag{C.5}$$

again by $\delta_n$-localizability. Finally, by the same reasoning, for the third term,

$$\mathbb{E}\Big[\Big(\delta^4 \sum_\ell \langle \nabla^2 u, \nabla H^\ell \otimes \nabla H^\ell - V \rangle_{\ell-1}^2\Big)^2\Big] \lesssim \delta^6(t-s)^2 \sup_{x\in E_{K,n}^*} ||\nabla^2 u(x)||^4 \cdot \mathbb{E}\big[||\nabla H(x)||^4\big]^2$$

$$\lesssim_K (t-s)^2. \tag{C.6}$$

All of the above terms are $O((t-s)^2)$ since $0 \le s, t \le T$. Thus we have the claimed (C.2), and by Kolmogorov's continuity theorem, $(\mathbf{v}_n(s \wedge \tau_K))_s$, are uniformly $1/4$-Holder and thus the sequence is tight with $1/4$-Holder limit points. Notice furthermore that if we look at $(\mathbf{v}_n(t \wedge \tau_K) - \mathbf{a}_n(t \wedge \tau_K))_t$, this sequence is also tight and the limits points are continuous martingales. Let us examine their limiting quadratic variations.

Let $\mathbf{v}_n^K(t) = \mathbf{v}_n(t \wedge \tau_K)$ and define $\mathbf{a}_n^K(t)$ and $\mathbf{b}_n^K(t)$ analogously. Furthermore, let $\mathbf{v}^K(t)$, $\mathbf{a}^K(t)$ and $\mathbf{b}^K(t)$ be their respective limits which we have established to exist and be $1/4$-Holder.

We will compute the limiting quadratic variation for $\mathbf{b}^K(t)$. For ease of notation, let $\Delta M_\ell^{u_i} = M_\ell^{u_i} - M_{\ell-1}^{u_i}$ and $\Delta \mathcal{E}_\ell^{u_i} = \mathcal{E}_\ell^{u_i} - \mathcal{E}_{\ell-1}^{u_i}$.

Notice first that for $1 \le i, j \le k$,

$$b_{n,i}^K(t) b_{n,j}^K(t) - \int_0^t \delta\mathbb{E}\big[\Delta M_{[s/\delta]\wedge\tau_K}^{u_i} \Delta M_{[s/\delta]\wedge\tau_K}^{u_j}\big]ds,$$

is a martingale. We therefore need to consider the limit as $n \to \infty$ of the integral above. We can write

$$\mathbb{E}[\Delta M_\ell^{u_i} \Delta M_\ell^{u_j}] = \langle \nabla u_i, V \nabla u_j \rangle + \delta\mathbb{E}[\langle \nabla H^\ell, \nabla u_i \rangle_{\ell-1} \Delta \mathcal{E}_\ell^{u_j}] + \delta\mathbb{E}[\langle \nabla H^\ell, \nabla u_j \rangle_{\ell-1} \Delta \mathcal{E}_\ell^{u_i}]$$
$$+ \delta^2 \mathbb{E}[\Delta \mathcal{E}_\ell^{u_i} \Delta \mathcal{E}_\ell^{u_j}].$$

Consider the integrals of $\delta$ times each of these four terms separately.

By the assumption of (2.2),

$$\sup_{t\le T} \Big| \int_0^t \delta \langle \nabla u_i, V \nabla u_j \rangle_{[s/\delta]\wedge\tau_K} ds - \int_0^t \Sigma_{ij}(\mathbf{v}_n^K(s))ds \Big|$$

$$\le \sup_{x\in E_{K,n}^*} |\delta \langle \nabla u_i, V \nabla u_j \rangle(x) - \Sigma_{ij}(\mathbf{u}_n(x))|,$$

goes to zero as $n \to \infty$.

We now reason that the integrals of the other three terms all go to zero as $n \to \infty$. We will show this for one of them, the arguments for the others being similar. By Cauchy–Schwarz,

$$\sup_{x\in E_{K,n}^*} |\delta^2\mathbb{E}[\langle \nabla H, \nabla u_i \rangle \Delta \mathcal{E}_\ell^{u_j}]| \le \delta^2\mathbb{E}[\langle \nabla H, \nabla u_i \rangle^2]^{1/2}\mathbb{E}[(\Delta \mathcal{E}_\ell^{u_i})^2]^{1/2}.$$

The first expectation contributes $\delta^{-1/2}$ by the first part of Item 3 of localizability. At the same time,

$$\mathbb{E}[(\Delta \mathcal{E}_\ell^{u_i})^2]^{1/2} \lesssim \mathbb{E}[\langle \nabla^2 u_i, \nabla \Phi \otimes \nabla H \rangle^2]^{1/2} + \mathbb{E}[\langle \nabla^2 u_i, \nabla H \otimes \nabla H - V \rangle^2]^{1/2}.$$

The first of these terms is at most $\delta^{-1}$ as argued in (C.5). The second is $o(\delta^{-3/2})$ by the second part of Item 3 in the definition of localizability. As such, we are able to conclude that

$$\sup_{t \le T} \left| \int_0^t \delta^2 \mathbb{E}[\langle \nabla H, \nabla u_i \rangle_{[s/\delta] \wedge \tau_K} \Delta \mathcal{E}_{[s/\delta] \wedge \tau_K}^{u_j}] ds \right|,$$

goes to zero as $n \to \infty$. Altogether, with the analogous reasoning for the other two terms, we conclude that

$$\lim_{n \to \infty} \sup_{i,j \le k} \sup_{t \le T} \left| \int_0^t \delta \mathbb{E}[\Delta M_{[s/\delta] \wedge \tau_K}^{u_i} \Delta M_{[s/\delta] \wedge \tau_K}^{u_j}] ds - \int_0^t \Sigma_{ij}(\mathbf{v}_n^K(s)) ds \right| = 0.$$

Thus, if we consider the continuous martingales given by $\mathbf{b}^K(t)$, its angle bracket is, by definition, given by

$$\langle \mathbf{b}^K \rangle_t = \int_0^t \mathbf{\Sigma}(\mathbf{v}^K(s)) ds.$$

By Ito's formula for continuous martingales (see, e.g., [18, Theorem 5.2.9]), we have that $f(\mathbf{v}_t) - \int_0^t \mathsf{L} f ds$ is a martingale for all $f \in C_0^\infty(\mathbb{R}^k)$ (smooth compactly supported functions on $\mathbb{R}^k$), where

$$\mathsf{L} = \frac{1}{2} \sum_{ij=1}^k \Sigma_{ij} \partial_i \partial_j - \sum_{i=1}^k h_i \partial_i.$$

Since, by assumption, $\mathbf{h}, \sqrt{\Sigma}$ are locally lipschitz—and thus lipschitz on $E_K$—this property uniquely characterizes the solutions to (2.3) (see, e.g., [46, Theorem 6.3.4]). Thus $\mathbf{v}_K$ converges to the solution of (2.3) stopped at $\tau_K$. Thus by a standard localization argument [46, Lemmas 11.1.11-12], every limit point $\mathbf{v}(t)$ of $\mathbf{v}_n(t)$ solves the SDE (2.3) (using here that $E_K$ is an exhaustion by compact sets of $\mathbb{R}^k$). $\qquad \square$

# D  Deferred proofs from Section 3

## D.1  The effective dynamics for Matrix and Tensor PCA

Our aim in this section is to establish Proposition 3.1, showing that the summary statistics $\mathbf{u}_n = (m, r_\perp^2)$ satisfy the conditions of Theorem 2.2 with the desired $\mathbf{f}, \mathbf{g}$ and $\Sigma$. In what follows, for ease of notation we will denote $r^2 = r_\perp^2$ and $R^2 = m^2 + r^2$. We first establish that the sequence $\mathbf{u}_n$ is $\delta_n$-localizable for any $\delta_n = O(1/n)$. The localizing sequence $E_K$ will simply be centered balls of radius $K$ in $\mathbb{R}^2$, say. We first check the regularity of the observable pair $\mathbf{u}_n$; express the Jacobian for that pair as

$$\nabla m = v, \qquad \nabla r^2 = 2(x - mv). \tag{D.1}$$

To check the regularity of observables, notice that $\nabla^2 m = 0$, while $\nabla^2 r^2 = 2(I - vv^T)$, whose operator norm is simply 2, and $\nabla^\ell u_i = 0$ for all $\ell \ge 3$. Next, we verify the regularity of the loss. In this appendix we will do things in the more general setting where we add a ridge penalty to the loss, so that for $\alpha > 0$ fixed, the loss is given by

$$L(x, Y) = -2(\langle W, x^{\otimes k} \rangle + \lambda \langle x, v \rangle^k) + ||x||^{2k} + \frac{\alpha}{2}||x||^2 + c(Y),$$

and thus $H(x) = -2\langle W, x^{\otimes k} \rangle$. In the coordinates $(m, r_\perp^2)$, we have $\Phi(x) = -2\lambda m^k + (r_\perp^2 + m^2)^k + \frac{\alpha}{2}(r_\perp^2 + m^2) + c'$. Observe that

$$\nabla \Phi = \partial_1 \phi \nabla m + \partial_2 \phi \nabla r^2.$$

where

$$\partial_1 \phi = -2\lambda k m^{k-1} + (2k R^{2k-2} + \alpha)m \qquad \partial_2 \phi = k R^{2k-2} + \frac{\alpha}{2}.$$

Notice that $\langle\nabla m,\nabla m\rangle=1, \langle\nabla m,\nabla r^2\rangle=0$, and $\langle\nabla r^2,\nabla r^2\rangle=4r^2$. Consider $\|\nabla\Phi\|\leq$ $|\partial_1\phi|\|\nabla m\|+|\partial_2\phi|\|\nabla r^2\|$; the bounding quantity is evidently a continuous function of $m,r^2$ and therefore as long as $x$ is such that $(m,r^2)\in E_K$, it is bounded by some $C(K)$. Next, if we consider
$$\mathbb{E}[\|\nabla H\|^4]\leq C_k\mathbb{E}[\|W(x,\ldots,x,\cdot)\|^4]\leq\mathbb{E}\|W\|_{op}^4\cdot R^{4k}\leq C(k,K)n^2$$
where the bound on the operator norm of an i.i.d. Gaussian $k$-tensor can be found, e.g., in [5]. By the same reasoning, for every $w$,
$$\mathbb{E}[\langle\nabla H,w\rangle^4]\leq 16k\mathbb{E}[|W(w,x,\ldots,x)|^4]\leq C(k,K)n^2\|w\|.$$

If $w=\nabla m=v$ then $\|w\|=1$ and if $w=\nabla r^2=2(x-mv)$ then $\|w\|\leq C(K)$, so in both cases this is at most $C(k,K)n^2$. Finally, $\nabla^2 u$ is only non-zero if $u=r$ in which case it is $I-vv^T$. Then,
$$\mathbb{E}[\langle\nabla^2 r,\nabla H\otimes\nabla H-V\rangle^2]\leq 2\mathbb{E}[\|\nabla H\|^2]\leq C(k,K)n$$

by the second item in the definition of localizability, and evidently the right-hand side is $o(\delta^{-2})$ if $\delta_n=O(1/n)$. This concludes the proof of $\delta_n$ localizability for every $\delta_n=O(1/n)$.

We now turn to calculating $\mathbf{f},\mathbf{g},\Sigma$. Starting with $\mathbf{f}$, by the above,
$$f_m=\langle\nabla\Phi,\nabla m\rangle=-2\lambda km^{k-1}+(2kR^{2k-2}+\alpha)m$$
$$f_{r^2}=\langle\nabla\Phi,\nabla r^2\rangle=2r^2(2kR^{2k-2}+\alpha).$$

We next turn to calculating the corrector. For this, we first calculate the matrix $V=\mathbb{E}[\nabla H\otimes\nabla H]$. Recalling that $H=-2\langle W,x^{\otimes k}\rangle$ where $W$ is an i.i.d. Gaussian $k$-tensor, we have that
$$V_{ij}=\mathbb{E}[\partial_i H\partial_j H]=4k(k-1)x_i x_j R^{2k-4}+\begin{cases}4kR^{2k-2} & i=j\\ 0 & i\neq j\end{cases}. \tag{D.2}$$

In particular, for $\delta=c_\delta/n$, we have
$$\delta\mathcal{L}^\delta m=0$$
$$\delta\mathcal{L}^\delta r^2=\frac{4c_\delta}{n}\sum_i(1-v_i^2)R^{2k-2}+\frac{4c_\delta}{n}k(k-1)r^2 R^{2k-4}$$
$$=\frac{4c_\delta}{n}k\Big((n-1)R^{2k-2}+(k-1)r^2 R^{2k-4}\Big)$$

from which we obtain in the limit that $n\to\infty$ that $g_m=0$ and $g_{r^2}=4c_\delta kR^{2k-2}$.

Together, these yield the ODE system of (3.1),
$$\dot u_1=2u_1(\lambda ku_1^{k-2}-kR^{2k-2}-\alpha),\qquad\dot u_2=-(4u_2-4c_\delta)kR^{2k-2}-2\alpha u_2.$$

which reduces in the $\alpha=0$ case to that claimed in Proposition 3.1.

Finally, in order to see that $\Sigma=0$, consider
$$JVJ^T=\begin{pmatrix}4k(k-1)m^2 R^{2k-4}+4kR^{2k-2} & 4k(k-1)m(R^2-m)R^{2k-4}\\ 4k(k-1)m(R^2-m)R^{2k-4} & 4k(k-1)(R^2-m)^2 R^{2k-4}\end{pmatrix}, \tag{D.3}$$

which when multiplied by $\delta=O(1/n)$ evidently vanishes.

## D.2 The fixed points of Proposition 3.1

We now turn to analyzing the ODE of Proposition 3.1 and obtaining the fixed point classification of Proposition 3.2. At the fixed points, we must have that
$$\lambda ku_1^{k-1}=\left(kR^{2k-2}+\alpha\right)u_1,$$
$$2c_\delta kR^{2k-2}=\left(2kR^{2k-2}+\alpha\right)u_2.$$

If $u_1=0$, then $R^2=u_2$ and there are two possible fixed points: either $u_2=0$ or $u_2$ solves
$$ku_2^{k-2}(2c_\delta-2u_2)=\alpha.$$

Notice that if $k = 2$, this has a nontrivial solution of the form $c_\delta - \frac{\alpha}{2} = u_2$, provided $\alpha < \alpha_c(2) := 2c_\delta$, and if $k > 2$, this has a nontrivial solution provided

$$\alpha \leq \max_{x \geq 0} kx^{k-2}(2c_\delta - 2x),$$

which is attained at $c_\delta(k-2)x^{k-3} - (k-1)x^{k-2} = 0$ which is at $\frac{c_\delta(k-2)}{k-1} = x$, which gives

$$\alpha < \alpha_c(k) := 2c_\delta^{k-1}k(k-1)^{-(k-1)}(k-2)^{k-2}.$$

Evidently when we take $\alpha = 0$, then its non-trivial solution is at $u_2 = 1$ for all $k \geq 2$.

Alternatively, if $u_1 \neq 0$ at a fixed point, then we can simplify further by dividing out by $u_1$ to get

$$\lambda u_1^{k-2} = R^{2k-2} + \frac{\alpha}{k}, \qquad \text{and} \qquad kR^{2k-2} = (kR^{2k-2} + \alpha)u_2,$$

so that at the fixed point,

$$u_1^{k-2} = \left(\frac{kR^{2k-2} + \alpha}{\lambda k}\right), \qquad \text{and} \qquad u_2 = \frac{2c_\delta kR^{2k-2}}{2kR^{2k-2} + \alpha}.$$

Let us for simplicity of calculations at this point set $\alpha = 0$ as is the case in Proposition 3.1. Then, we simply get $u_2 = c_\delta$. In the case of $k = 2$, we also find that there is a solution if and only if $\lambda > c_\delta$, in which case $R^2 = \lambda$, from which together with $R^2 = u_1^2 + u_2$, we also get $u_1 = \pm\sqrt{\lambda - c_\delta}$.

In the general case of $k > 2$, we find that

$$R^2 = c_\delta + \lambda^{-\frac{2}{k-2}}R^{\frac{4(k-1)}{k-2}}.$$

This has real solutions (all of which have $R \geq u_2 = c_\delta$ as required) whenever $\lambda > \lambda_c(k)$ defined as

$$\lambda_c(k) := \left(\frac{c_\delta}{k}\right)^{k/2}\left(\frac{(2k-2)^{k-1}}{(k-2)^{(k-2)/2}}\right). \tag{D.4}$$

(Notice that with the interpretation $0^0 = 1$, this returns $\lambda_c(2) = c_\delta$.) With this choice of $\lambda$, then, whenever $\lambda > \lambda_c(k)$, the equation for $R^2$ has exactly two real solutions, both of which are at least $c_\delta$ which we can denote by

$$\rho_\dagger(k, \lambda) := \inf\{\rho \geq 1 : \lambda^{-\frac{2}{k-2}}\rho^{\frac{2(k-1)}{k-2}} - \rho + c_\delta = 0\},$$

$$\rho_\star(k, \lambda) := \sup\{\rho \geq 1 : \lambda^{-\frac{2}{k-2}}\rho^{\frac{2(k-1)}{k-2}} - \rho + c_\delta = 0\}.$$

When $\lambda > \lambda_c(k)$, $\rho_\dagger < \rho_\star$ and when $\lambda = \lambda_c(k)$, the two are equal. Given this, we can then solve for $\tilde{u}_1$ at the corresponding fixed point, and find that they occur at

$$m_\dagger(k, \lambda) = \sqrt{\rho_\dagger - c_\delta}, \qquad \text{and} \qquad m_\star(k, \lambda) = \sqrt{\rho_\star - c_\delta}. \tag{D.5}$$

### D.3 Effective dynamics for the population loss

In practice it is often most useful to track the loss, or ideally, the generalization error. In this subsection, we add the generalization error $\Phi$ to our set of summary statistics and obtain limiting equations for its evolution. For simplicity of calculations let us stick to $\alpha = 0$.

$$f_\Phi = \langle \nabla\Phi, \nabla\Phi \rangle = 4\lambda^2 k^2 m^{2(k-1)} - 8\lambda k^2 m^k R^{2k-2} + 4k^2 R^{4k-4}m^2 + 4k^2 r^2 R^{4k-4}$$

$$= 4k^2 m^2\left(\lambda^2 m^{2(k-2)} - 2\lambda m^{k-2}R^{2k-2} + R^{4k-4}\right) + 4k^2 r^2 R^{4k-4}.$$

Next, consider the corrector for $\Phi$. For this, notice that

$$\tfrac{1}{2}\nabla^2\Phi = -\lambda k(k-1)m^{k-2}\nabla m^{\otimes 2} + kR^{2k-2}\nabla m^{\otimes 2} + k(k-1)R^{2(k-2)}(2m\nabla m + \nabla r^2) \otimes \nabla m$$

$$+ k(k-1)R^{2(k-2)}(2m\nabla m \otimes \nabla r^2 + \nabla r^2 \otimes \nabla r^2) + \tfrac{1}{2}\partial_2\phi\nabla^2 r^2.$$

Recalling $V$ from (D.2), and taking $\delta = c_\delta/n$, all the terms in $\sum_{ij} V_{ij}\partial_i\partial_j\Phi$ vanish in the limit except the contribution from the $\nabla^2 r^2$, which yields

$$g_\Phi = \lim_{n\to\infty} \delta\mathcal{L}^\delta\Phi = 4c_\delta k^2 R^{4(k-1)}$$

Finally, we wish to compute the volatility for the stochastic part of the evolution of $\Phi$. For this, consider $\nabla\Phi V \nabla\Phi^T$ and notice that all the entries of that matrix are continuous functions of $\mathbf{u}_n$ and therefore when multiplied by $\delta = O(1/n)$, the limit as $n \to \infty$ of $\Sigma$ vanishes. We are left with

$$\dot{\Phi} = -4k^2 m^2\left(\lambda^2 m^{2(k-2)} - 2\lambda m^{k-2}R^{2k-2} + R^{4k-4}\right) - 4k^2 R^{4(k-1)}(r^2 - c_\delta). \tag{D.6}$$

One could then perform the fixed point analysis directly on (D.6) if desired.

### D.4 Diffusive limits at the equator

In this subsection, we develop the stochastic limit theorems for the rescaled observables about the axis $m = 0$. Here we take as variables $(\tilde{u}_1, \tilde{u}_2) = (\sqrt{n}m, r^2)$. For simplicity of presentation, we take $\alpha = 0$ and $c_\delta = 1$ here. In this case, the change from the previous pair of variables is in the $J$ matrix, in which now $\nabla \tilde{u}_1 = \sqrt{n}\nabla m = \sqrt{n}v$. As such,

$$\langle \nabla \Phi, \nabla \tilde{u}_1 \rangle = -2k\lambda\sqrt{n}m^{k-1} + 2k\sqrt{n}R^{2k-2}m = -2k\lambda n^{-\frac{k-2}{2}}\tilde{u}_1^{k-1} + 2k(r^2 + (\tilde{u}_1^2/n))^{k-1}\tilde{u}_1,$$

$$\langle \nabla \Phi, \nabla r^2 \rangle = 4kr^2R^{2k-2} = 4kr^2(r^2 + (\tilde{u}_1^2/n))^{k-1}.$$

Taking limits as $n \to \infty$, as long as $\lambda$ is fixed in $n$, we see that $\mathbf{f}$ is given by

$$f_{\tilde{u}_1} = \begin{cases} -2\lambda\tilde{u}_1^{k-1} + 2k\tilde{u}_2^{k-1}\tilde{u}_1 & k = 2 \\ 2k\tilde{u}_2^{k-1}\tilde{u}_1 & k \geq 3 \end{cases}, \quad \text{and} \quad f_{\tilde{u}_2} = 4k\tilde{u}_2^k.$$

We turn to obtaining the correctors in these rescaled coordinates. Evidently $\delta\mathcal{L}\tilde{u}_1 = 0$ still by linearity of $\tilde{u}_1$. Following the calculation for the corrector, we find that it is now given by $g_{\tilde{u}_2} = 4k\tilde{u}_2^{k-1}$.

Next we consider the volatility of the stochastic process one gets in the limit. Recalling $JVJ^T$ from (D.3), and noticing that the rescaling $J \to \tilde{J}$ multiplies its $(1,1)$-entry by $n$ and its off-diagonal entries by $\sqrt{n}$, we find that in the new coordinates,

$$\tilde{J}V\tilde{J}^T = \begin{pmatrix} 4k(k-1)\tilde{u}_1^2 R^{2k-4} + 4knR^{2k-2} & 4k(k-1)\tilde{u}_1(R^2 - m)R^{2k-4} \\ 4k(k-1)\tilde{u}_1(R^2 - m)R^{2k-4} & 4k(k-1)(R^2 - m)^2 R^{2k-4} \end{pmatrix} \tag{D.7}$$

Multiplying by $\delta = 1/n$ and taking the limit as $n \to \infty$, the only entry of this matrix that survives is from $\Sigma_{11}$ where we get $\Sigma_{11} = 4k\tilde{u}_2^{k-1}$. Putting the above together yields the claimed Proposition 3.3.

Regarding the discussion in the $k \geq 3$ case when $\lambda_n = \Lambda n^{(k-2)/2}$, observe that the first term in $\langle \Phi, \nabla \tilde{u}_1 \rangle$ above would not vanish and would instead converge to $-4k\Lambda\tilde{u}_1^{k-1}$.

## E  Deferred proofs from Section 4

### E.1  The summary statistics

Recall the cross-entropy loss for the binary GMM with SGD from (4.1), and recall the set of summary statistics $\mathbf{u}_n$ from (4.2). The next lemma shows that $\mathbf{u}_n$ form a good set of summary statistics.

**Lemma E.1.** *The distribution of $L((v, W))$ depends only on $\mathbf{u}_n$ from (4.2). In particular, we have that $\Phi(x) = \phi(\mathbf{u}_n)$ for some $\phi$. Furthermore, $\mathbf{u}_n$ satisfy the bounds in item (1) of Definition 2.1 if $E_K$ is the ball of radius $K$ in $\mathbb{R}^{2N+2}$.*

*Proof.* Let $X_\mu \sim \mathcal{N}(\mu, I/\lambda)$ and $X_{-\mu} \sim \mathcal{N}(-\mu, I/\lambda)$. Then, notice that

$$L((v, W)) \stackrel{d}{=} \begin{cases} -v \cdot g(WX_\mu) + \log(1 + e^{v \cdot g(WX_\mu)}) + p(v, W) & \text{w. prob. } 1/2 \\ \log(1 + e^{v \cdot g(-WX_\mu)}) + p(v, W) & \text{w. prob. } 1/2 \end{cases}.$$

Next, notice that as a vector,

$$(W_1 X_\mu, W_2 X_\mu) \stackrel{d}{=} (m_1 + Z_{1,\mu}m_1 + Z_{1,\perp}, m_2 + Z_{2,\mu}m_2 + Z_{2,\perp})$$

where $Z_{1,\mu}, Z_{2,\mu}$ are i.i.d. $\mathcal{N}(0, \lambda^{-1})$, and $Z_{1,\perp}, Z_{2,\perp}$ are jointly Gaussian with means zero and covariance

$$\lambda^{-1} \begin{bmatrix} R_{11}^\perp & R_{12}^\perp \\ R_{12}^\perp & R_{22}^\perp \end{bmatrix} \tag{E.1}$$

Similarly, the distribution of $WX_{-\mu}$ also only depends on $(m_i, R_{ij}^\perp)_{i,j}$. Finally,

$$p(v, W) = \frac{\alpha}{2}(v_1^2 + v_2^2 + m_1^2 + R_{11}^\perp + m_2^2 + R_{22}^\perp)$$

Therefore, at a fixed point, the law of $L((v, W))$ is simply a function of $\mathbf{u}_n(v, W)$. This of course implies the same for the population loss $\Phi$.

To see that the summary statistics satisfy the bounds of item (1) in Definition 2.1, write $\nabla = (\partial_{v_1}, \partial_{v_2}, \nabla_{W_1}, \nabla_{W_2})$. Then

$$
J = (\nabla u_\ell)_\ell = \begin{bmatrix}
(1, 0, 0, 0) \\
(0, 1, 0, 0) \\
(0, 0, \mu, 0) \\
(0, 0, 0, \mu) \\
(0, 0, W_2^\perp, W_1^\perp) \\
(0, 0, 2W_1^\perp, 0) \\
(0, 0, 0, 2W_2^\perp)
\end{bmatrix}
\tag{E.2}
$$

For the higher derivatives, evidently we only have second derivatives in the last 3 variables each of which is given by a block diagonal matrix where only one block is non-zero and is given by an identity matrix. The third derivatives of all elements of $\mathbf{u}_n$ are zero. $\qquad\square$

We can now express the loss, the population loss, and their respective derivatives and they (their laws at a fixed point) will evidently only depend on the summary statistics. One arrives at the following expressions for $\nabla L$ by direct calculation from (4.1).

$$
\nabla_{v_i} L = (W_i \cdot X) \mathbf{1}_{W_i \cdot X \geq 0} \big( -y + \sigma(v \cdot g(WX)) \big) + \alpha v_i
\tag{E.3}
$$
$$
\nabla_{W_i} L = v_i X \mathbf{1}_{W_i \cdot X \geq 0} \big( -y + \sigma(v \cdot g(WX)) \big) + \alpha W_i
\tag{E.4}
$$

In what follows, for an arbitrary vector $w \in \mathbb{R}^N$, we use the notation

$$
\mathbf{A}_i = \mathbb{E}\big[ X \mathbf{1}_{W_i \cdot X \geq 0} \big( -y + \sigma(v \cdot g(WX)) \big) \big]
\tag{E.5}
$$

(Notice that if $w \in \{\mu, W_i, W_i^\perp\}$, then $\mathbf{A}_i \cdot w$ is only a function of $\mathbf{u}_n$ by the same reasoning as used in Lemma E.1.) Then, we can also easily express

$$
\nabla_{v_i} \Phi = W_i \cdot \mathbf{A}_i + \alpha v_i
\tag{E.6}
$$
$$
\nabla_{W_i} \Phi = v_i \mathbf{A}_i + \alpha W_i
\tag{E.7}
$$

and for $H = L - \Phi$,

$$
\nabla_{v_i} H = W_i \cdot \Big( X \mathbf{1}_{W_i \cdot X \geq 0} \big( -y + \sigma(v \cdot g(WX)) \big) - \mathbf{A}_i \Big),
\tag{E.8}
$$
$$
\nabla_{W_i} H = v_i \Big( X \mathbf{1}_{W_i \cdot X \geq 0} \big( -y + \sigma(v \cdot g(WX)) \big) - \mathbf{A}_i \Big).
\tag{E.9}
$$

Finally, the matrix $V$ can be expressed as follows:

$$
V_{v_i, v_j} = \mathbb{E}\big[ (W_i \cdot X)(W_j \cdot X) \mathbf{1}_{W_i \cdot X \geq 0} \mathbf{1}_{W_j \cdot X \geq 0} (-y + \sigma(v \cdot g(WX)))^2 \big] - (W_i \cdot \mathbf{A}_i)(W_j \cdot \mathbf{A}_j)
$$
$$
V_{v_i, W_j} = v_j \mathbb{E}\big[ (W_i \cdot X) X \mathbf{1}_{W_i \cdot X \geq 0} \mathbf{1}_{W_j \cdot X \geq 0} (-y + \sigma(v \cdot g(WX)))^2 \big] - v_j (W_i \cdot \mathbf{A}_i) \mathbf{A}_j
$$
$$
V_{W_i, W_j} = v_i v_j \mathbb{E}\big[ X^{\otimes 2} \mathbf{1}_{W_i \cdot X \geq 0} \mathbf{1}_{W_j \cdot X \geq 0} (-y + \sigma(v \cdot g(WX)))^2 \big] - v_i v_j \mathbf{A}_i \otimes \mathbf{A}_j.
\tag{E.10}
$$

Let us conclude this subsection with the following simple preliminary bound that will be useful towards establishing the conditions of $\delta_n$-localizability from Definition 2.1.

**Lemma E.2.** *For every fixed $w \in \mathbb{R}^n$, we have*

$$
\mathbb{E}[|X \cdot w|^4] \lesssim (w \cdot \mu)^4 + \|w\|^4 \lambda^{-2}, \qquad \text{and} \qquad \|\mathbf{A}_i\| \leq C(\mathbf{u}_n).
$$

*Proof.* For the first bound, let $Z \sim \mathcal{N}(0, I)$ and consider

$$
\mathbb{E}[|X \cdot w|^4] = \frac{1}{2} \mathbb{E}[(w \cdot \mu + \lambda^{-1/2} w \cdot Z)^4] + \frac{1}{2} \mathbb{E}[(-w \cdot \mu + \lambda^{-1/2} w \cdot Z)^4].
$$

Using the fact that $Z$ is mean zero, and pulling out $w \cdot \mu$, we see that this is at most some universal constant times

$$
(w \cdot \mu)^4 + \lambda^{-2} \mathbb{E}[(w \cdot Z)^4].
$$

For the second term, notice that $w \cdot Z$ is distributed as $z \sim \mathcal{N}(0, \|w\|^2)$, implying the desired.

The bound on $\mathbf{A}_i$ goes as follows. Evidently it suffices to let $X_\mu = \mu + \lambda^{-1/2}Z$ for $Z \sim \mathcal{N}(0, I)$, and prove the bound on the norm of

$$\mathbb{E}[X_\mu \mathbf{1}_{W_i \cdot X_\mu \geq 0}(-1 + \sigma(g(WX_\mu)))] = \mathbb{E}[(\mu + \lambda^{-1/2}Z)\mathbf{1}_{W_i \cdot X_\mu \geq 0}(-1 + \sigma(g(WX_\mu)))].$$

Now decompose $Z$ as

$$Z_\mu \mu + Z_{1,\perp}W_1^\perp + Z_{2,\perp}W_2^\perp + Z_3,$$

where $Z_\mu \sim \mathcal{N}(0,1)$ is independent of $(Z_{1,\perp}, Z_{2,\perp})$ which is distributed as $\mathcal{N}(0, A)$ with $A$ given by (E.1), which is independent of $Z_3$ distributed as a standard Gaussian vector orthogonal to the subspace spanned by $(\mu, W_1^\perp, W_2^\perp)$. By independence of $Z_3$ from the indicator and the argument of the sigmoid, all those terms contribute nothing to the expectation, and therefore,

$$\|\mathbf{A}_i\|^2 \leq \sum_{w \in \{\mu, W_1^\perp, W_2^\perp\}} \mathbb{E}[(X \cdot w)^2 \mathbf{1}_{W_i \cdot X \geq 0}(-y + \sigma(g(WX)))] \leq (1 + R_{11}^\perp + R_{22}^\perp)(1 + \lambda^{-1}).$$

Here, we used the first inequality of the lemma. This yields the desired. $\qquad\square$

## E.2   Verifying the conditions of Theorem 2.2 for fixed $\lambda$

Throughout this section we will take $\mu = e_1$. By rotational invariance of the problem, this is without loss of generality, and only simplifies certain expressions.

**Lemma E.3.** *For $\delta_n = O(1/N)$ and any fixed $\lambda$, the 2-layer GMM with observables $\mathbf{u}_n$ is $\delta_n$-localizable for $E_K$ being balls of radius $K$ about the origin in $\mathbb{R}^7$.*

*Proof.* The condition on $\mathbf{u}_n$ was satisfied per Lemma E.1. Recalling $\nabla\Phi$ from (E.6)–(E.7), one can verify that the norm of each of the four terms in $\nabla\Phi$ is individually bounded, using the Cauchy–Schwarz inequality together with the bound of Lemma E.2 on $\|\mathbf{A}_i\|$.

Next, consider bounding $\mathbb{E}[\|\nabla H\|^4]$ by

$$\mathbb{E}[\|\nabla H\|^4] \leq \sum_{i=1,2} \mathbb{E}[|\nabla_{v_i}H|^4] + \mathbb{E}[\|\nabla_{W_i}H\|^4],$$

and recall the expressions for $\nabla H$ from (E.8)–(E.9). Using the trivial bound $|\sigma(x)| \leq 1$, and the inequality $(a+b)^4 \leq C(a^4 + b^4)$, for $i \in \{1,2\}$, the first term is at most

$$C\big(\mathbb{E}[|X \cdot W_i|^4] + \|W_i\|^4 \|\mathbf{A}_i\|^4\big),$$

which is bounded by a constant depending continuously on $\mathbf{u}_n$ per Lemma E.2. If we let $Z$ be a standard Gaussian, the second term is evidently governed by

$$C\Big(v_i^4 \mathbb{E}\Big[\|X\mathbf{1}_{W_i \cdot X \geq 0}\sigma(-v \cdot g(WX))\|^4\Big] + v_i^4 \|\mathbf{A}_i\|^4\Big) \leq C|v_i|^4\Big(1 + \frac{\mathbb{E}\|Z\|^4}{\lambda^2}\Big).$$

Using the well-known bound that $\mathbb{E}[\|Z\|^4] \leq N^2$, and the fact that $\delta = O(1/N)$, we see that this is at most $C\delta^{-2}$ as needed.

We turn to the third item in the definition of localizability. We next verify the claimed bound that

$$\delta_n^2 \sup_i \sup_{x \in \mathbf{u}_n^{-1}(E_K)} \mathbb{E}[\langle \nabla H, \nabla u_i\rangle^4] \leq C(K). \tag{E.11}$$

When $u_i$ is $v_i$, this is simply a fourth moment bound on $\nabla_{v_i}H$, which follows as the third moment bound did, with no need for the $\delta_n^2$. When $u_i$ is $m_i$, or $R_{ij}^\perp$, the bound follows from

$$\mathbb{E}[\langle \nabla_{W_i}H, w\rangle^4] \leq C|v_i|^4\big(\mathbb{E}[|X \cdot w|^4] + \|w\|^4 \|\mathbf{A}_i\|^4\big),$$

for choices of $w$ being either $\mu$ in which case $\|w\| = 1$ or $W_i^\perp$ in which case $\|w\| = R_{ii}^\perp$. For each $K$, this is at most some constant $C(K)$ using the two bounds of Lemma E.2. Again, we note that the factor of $\delta_n^2$ wasn't needed.

Finally, consider the quantity

$$\mathbb{E}[\langle \nabla^2 u, \nabla H \otimes \nabla H - V\rangle^2]$$

This is only non-zero for $u \in \{R_{ij}^\perp\}$ for which $\nabla^2 u$ is a block-identity matrix, having operator norm at most 2 in all cases. Therefore, this quantity is at most $4\mathbb{E}[\|\nabla H\|^2]$ which is at most $N$ by the above proved second item in the definition of localizability. This is therefore $O(\delta_n^{-1}) = o(\delta_n^{-2})$ as needed. $\qquad\square$

**Proof of Proposition 4.1.** The convergence of the population drift to $\mathbf{f}$ from Proposition 4.1 follows by taking the inner products of $\nabla L$ from (E.6) with the rows of $J$ from (E.2), and noticing that $\mathbf{A}_i^\mu$ from (4.3) is exactly $\mathbf{A}_i \cdot \mu$ and $\mathbf{A}_{ij}^\perp$ from (4.3) is exactly $\mathbf{A}_i \cdot W_j^\perp$.

Next consider the convergence of the correctors to the claimed $\mathbf{g}$. The variables $u \in \{v_1, v_2, m_1, m_2\}$ are linear so $\mathcal{L}_n u = 0$ and for these, $\mathbf{g}_u = 0$. For $u = R_{ij}^\perp$ for $i, j \in \{1, 2\}$, the relevant entries in $V$ are those corresponding to $W_i^\perp$ and $W_j^\perp$. For ease of notation, in what follows let $\pi = \sigma(v \cdot g(WX))$. For ease of calculation taking $\mu = e_1$, we have

$$\mathcal{L}_n R_{ij}^\perp = \sum_{k \neq 1} V_{W_{ik}, W_{jk}} \,,$$

which by (E.10), and the choice of $\delta_n = c_\delta/N$, is given by

$$\delta_n \mathcal{L}_n R_{ij}^\perp = \frac{c_\delta}{N} \sum_{k \neq 1} v_i v_j \Big( \mathbb{E}\big[(X \cdot e_k)^2 \mathbf{1}_{W_i \cdot X \geq 0} \mathbf{1}_{W_j \cdot X \geq 0} (-y + \pi)^2\big] - (\mathbf{A}_i \cdot e_k)(\mathbf{A}_j \cdot e_k) \Big)$$

$$= \frac{c_\delta}{N} v_i v_j \Big( \mathbb{E}\big[\|X^\perp\|^2 \mathbf{1}_{W_i \cdot X \geq 0} \mathbf{1}_{W_j \cdot X \geq 0} (-y + \pi)^2\big] - \langle \mathbf{A}_i - \mathbf{A}_i^\mu \mu, \mathbf{A}_j - \mathbf{A}_j^\mu \mu \rangle \Big) .$$

$$\text{(E.12)}$$

Let us first consider the two terms separately. For the first term, rewrite

$$\frac{1}{N} \mathbb{E}\big[\|X^\perp\|^2 \mathbf{1}_{W_i \cdot X \geq 0} \mathbf{1}_{W_j \cdot X \geq 0} (-y + \pi)^2\big]$$

$$= \mathbb{E}\big[\big(\tfrac{1}{N}\|X^\perp\|^2 - \lambda^{-1}\big) \mathbf{1}_{W_i \cdot X \geq 0} \mathbf{1}_{W_j \cdot X \geq 0} (-y + \pi)^2\big] + \lambda^{-1} \mathbf{B}_{ij} \,.$$

Of course the second term is exactly what we want to be $g_u$, so we will show the first term here goes to zero. By Cauchy–Schwarz, if $Z \sim \mathcal{N}(0, I - e_1^{\otimes 2})$, the first term above is at most

$$\lambda^{-1} \mathbb{E}\Big[\Big(\frac{\|Z\|^2}{N} - 1\Big)^2\Big]^{1/2} \leq \frac{2}{\lambda\sqrt{N}} \,,$$

where we used the fact that for a standard Gaussian, $g \sim \mathcal{N}(0, 1)$, we have $\mathbb{E}[(g^2 - 1)^2] = 2$. It remains to show the inner product term in (E.12) goes to zero as $n \to \infty$. For this term, rewrite

$$\frac{1}{N} \langle \mathbf{A}_i - \mathbf{A}_i^\mu \mu, \mathbf{A}_j - \mathbf{A}_j^\mu \mu \rangle = \frac{1}{N} \mathbb{E}\big[(X_1^\perp \cdot X_2^\perp) \mathbf{1}_{W_i \cdot X_1 \geq 0} \mathbf{1}_{W_j \cdot X_2 \geq 0} (-y + \pi_1)(-y + \pi_2)\big] \,,$$

where $X_1, X_2$ are i.i.d. copies of $X$, and $\pi_1, \pi_2$ are the corresponding $\sigma(v \cdot g(WX_1))$ and $\sigma(v \cdot g(WX_2))$. By Cauchy–Schwarz, if $Z, Z'$ are i.i.d. $\mathcal{N}(0, I - e_1^{\otimes 2})$, this is at most

$$\frac{1}{\lambda N} \mathbb{E}\big[(Z \cdot Z')^2\big]^{1/2} \leq \frac{1}{\lambda\sqrt{N}} \,.$$

This term therefore also vanishes as $n \to \infty$, yielding the desired limit for the corrector,

$$g_{R_{ij}^\perp} = \frac{c_\delta v_i v_j}{\lambda} \mathbb{E}\big[\mathbf{1}_{W_i \cdot X \geq 0} \mathbf{1}_{W_j \cdot X \geq 0} (-y + \pi)^2\big] = \frac{c_\delta v_i v_j}{\lambda} \mathbf{B}_{ij} \,.$$

which we emphasize is only a function of $\mathbf{u}_n$.

We lastly need to show that the diffusion matrix $\Sigma_n$ goes to zero as $n \to \infty$ when $\delta_n = O(1/n)$. This is straightforward to see by considering any element of $JVJ^T$ and using Cauchy–Schwarz together with the two bounds of Lemma E.2 to bound it in absolute value by some $C(K)$ independent of $n$. Then when multiplying by any $\delta_n = o(1)$, this entire matrix will evidently vanish. $\qquad\square$

### E.3 Preliminary estimate for small noise limits

Our next aim is to consider the effective dynamics of Proposition 4.1 in the small noise ($\lambda \to \infty$) limit. In this subsection, we collect some simple estimates that will make obtaining that limit easier. The first of these is the following elementary fact bounding the exponential moment of a Gaussian. As before, let $X_\mu \sim \mathcal{N}(\mu, I/\lambda)$.

**Fact E.1.** *Fix $\mu \in S^{N-1}(1)$, and let $g(x) = x \vee 0$. There is a function $C : \mathbb{R}^2 \to \mathbb{R}_+$ such that the following holds: for all $\lambda > 0$, $\theta \in \mathbb{R}$, and $(v_i, W_i) \in \mathbb{R} \times \mathbb{R}^N$,*

$$\mathbb{E}[\exp(\theta v_i g(W_i \cdot X_\mu))] \leq \exp\left(\theta v_i m_i + \tfrac{1}{2\lambda}\theta^2 v_i^2 R_{ii}^\perp\right).$$

The next lemma concerns the limits as $\lambda \to \infty$ of some of the building block terms we encounter.

**Lemma E.4.** *For each $i$, for every $R_{ii}^\perp < \infty$ and every $m_i > 0$, we have*

$$\lim_{\lambda \to \infty} \mathbb{P}\big(W_i \cdot X_\mu < 0\big) = 0. \tag{E.13}$$

*For every $v_i$, $R_{ij}^\perp$ and $m_i \neq 0$ for $i, j = 1, 2$, we have*

$$\lim_{\lambda \to \infty} \mathbb{E}\big[\big|\sigma(v \cdot g(WX_\mu)) - \sigma(v \cdot g(m))\big|\big] = 0. \tag{E.14}$$

*Proof.* The proof of (E.13) is easily seen by rewriting the probability in question as

$$\mathbb{P}(W_i \cdot X_\mu < 0) = \mathbb{P}\big(\mathcal{N}(0, \lambda^{-1}) < -m_i(m_i^2 + R_{ii}^\perp)^{-1/2}\big) = e^{-m_i^2 \lambda/2(m_i^2 + R_{ii}^\perp)},$$

so that as long as $m_i > 0$ this goes to zero as $\lambda \to \infty$.

We turn to (E.14). Consider

$$\mathbb{E}\big[\big|\sigma(v \cdot g(WX_\mu)) - \sigma(v \cdot g(m))\big|\big] \leq \mathbb{E}\Big[\big|e^{v \cdot g(WX_\mu)} - e^{v \cdot g(m)}\big|\Big]$$
$$\leq \mathbb{E}\big[\big|e^{v_1 g(W_1 \cdot X_\mu)} e^{v_2 g(W_2 \cdot X_\mu)} - e^{v_1 g(m_1)} e^{v_2 g(m_2)}\big|\big].$$

This in turn is bounded by

$$\mathbb{E}\big[e^{v_2 g(W_2 X_\mu)}\big|e^{v_1 g(W_1 X_\mu)} - e^{v_1 g(m_1)}\big|\big] + e^{v_1 g(m_1)}\mathbb{E}\big[\big|e^{v_2 g(W_2 X_\mu)} - e^{v_2 g(m_2)}\big|\big]. \tag{E.15}$$

Applying Cauchy–Schwarz to the first term, it suffices to establish the following bounds

$$\mathbb{E}\big[e^{2v_i g(W_i X_\mu)}\big] \leq C, \qquad \text{and} \qquad \lim_{\lambda \to \infty} \mathbb{E}\big[\big(e^{v_i g(W_i X_\mu)} - e^{v_i g(m_i)}\big)^2\big] = 0.$$

To demonstrate the first of these inequalities, notice that

$$\mathbb{E}\Big[e^{2v_i g(W_i X_\mu)}\Big] \leq \mathbb{E}\Big[e^{2v_i|W_i X_\mu|}\Big] \leq C.$$

uniformly over $\lambda$, per Fact E.1. For the second desired bound, expand $e^{v_i g(W_i \cdot X_\mu)} - e^{v_i g(m_i)}$ as

$$\big(e^{v_i(W_i \cdot X_\mu)\mathbf{1}_{W_i \cdot X_\mu \geq 0}} - e^{v_i(W_i \cdot X_\mu)\mathbf{1}_{m_i \geq 0}}\big) + \big(e^{v_i(W_i \cdot X_\mu)\mathbf{1}_{m_i \geq 0}} - e^{v_i m_i \mathbf{1}_{m_i \geq 0}}\big).$$

It suffices to show the expectation of the square of each of these goes to zero as $\lambda \to \infty$. First,

$$\mathbb{E}\big[\big(e^{v_i(W_i \cdot X_\mu)\mathbf{1}_{W_i \cdot X_\mu \geq 0}} - e^{v_i(W_i \cdot X_\mu)\mathbf{1}_{m_i \geq 0}}\big)^2\big] \leq (1 \vee e^{v_i(W_i \cdot X_\mu)})\mathbb{E}[\mathbf{1}_{W_i \cdot X_\mu \geq 0} - \mathbf{1}_{m_i \geq 0}].$$

If $m_i \neq 0$, the expectation on the right goes to zero by (E.13). Second,

$$\mathbb{E}\big[\big(e^{v_i(W_i \cdot X_\mu)\mathbf{1}_{m_i \geq 0}} - e^{v_i m_i \mathbf{1}_{m_i \geq 0}}\big)^2\big] \leq \mathbb{E}\big[(e^{v_i(W_i \cdot X_\mu)} - e^{v_i m_i})^2 \mathbf{1}_{m_i \geq 0}\big].$$

When $m_i < 0$, this is evidently zero; when $m_i > 0$, if $G_\lambda \sim \mathcal{N}(0, I/\lambda)$, this is

$$e^{2v_i m_i}\mathbb{E}\big[(e^{v_i(W_i \cdot G_\lambda)} - 1)^2\big].$$

which goes to zero as $O(\lambda^{-1})$ when $\lambda \to \infty$, by the explicit formula for the moment generating function of the Gaussian $W_i \cdot G_\lambda$, whose variance is $(m_i^2 + R_{ii}^\perp)\lambda^{-1}$. $\qquad\square$

### E.4 The small-noise limit of the effective dynamics

Let us consider the behavior of the ODE system of Proposition 4.1 in the limit that $\lambda \to \infty$.

**Proof of Proposition 4.2.** We begin with considering $\lim_{\lambda \to \infty} \mathbf{A}_i^\mu$: its limiting value will depend on the signs of both $m_1$ and $m_2$. We can express $\mathbf{A}_i^\mu$ from (4.3) as

$$
\mathbb{E}[(X \cdot \mu)\mathbf{1}_{W_i \cdot X \geq 0}(-y + \sigma(v \cdot g(WX)))] = \frac{1}{2}\mathbb{E}\Big[(X_\mu \cdot \mu)\mathbf{1}_{W_i \cdot X_\mu \geq 0}(-1 + \sigma(v \cdot g(WX_\mu)))\Big]
$$
$$
+ \frac{1}{2}\mathbb{E}\Big[(-X_\mu \cdot \mu)\mathbf{1}_{W_i \cdot X_\mu \leq 0}\sigma(v \cdot g(-WX_\mu))\Big].
$$

We claim that the two terms on the right-hand side converge to $-\frac{1}{2}\mathbf{1}_{m_i > 0}\sigma(-v \cdot g(m))$ and $-\frac{1}{2}\mathbf{1}_{m_i < 0}\sigma(v \cdot g(-m))$ respectively. This follows by e.g., writing the difference as

$$
\mathbb{E}\Big[(X_\mu \cdot \mu)\mathbf{1}_{W_i \cdot X_\mu \geq 0}\sigma(-v \cdot g(WX_\mu))\Big] - \mathbf{1}_{m_i \geq 0}\sigma(-v \cdot g(m)) \tag{E.16}
$$
$$
= \mathbb{E}\Big[(X_\mu \cdot \mu - 1)\mathbf{1}_{W_i \cdot X_\mu \geq 0}\sigma(-v \cdot g(WX_\mu))\Big]
$$
$$
+ \mathbb{E}\Big[(\mathbf{1}_{W_i \cdot X_\mu \geq 0} - \mathbf{1}_{m_i \geq 0})\sigma(-v \cdot g(WX_\mu))\Big]
$$
$$
+ \mathbf{1}_{m_i \geq 0}\mathbb{E}\Big[\sigma(-v \cdot g(WX_\mu)) - \sigma(-v \cdot g(m))\Big].
$$

Call these three terms $I, II$, and $III$. For $I$, we use the fact that $\mathbb{E}[|X_\mu \cdot \mu - 1|]$ goes to zero as $\lambda \to \infty$; $II$ is evidently bounded by $\mathbb{P}(W_i \cdot X_\mu < 0)$ when $m_i > 0$ or its symmetric counterpart when $m_i < 0$—both vanishing as $\lambda \to \infty$ per (E.13) in Lemma E.4; finally, $III$ goes to zero as $\lambda \to \infty$ by (E.14) in Lemma E.4.

Putting the above together, we find that

$$
\lim_{\lambda \to \infty} \mathbf{A}_i^\mu = -\frac{1}{2}\mathbf{1}_{m_i > 0}\sigma(-v \cdot g(m)) - \frac{1}{2}\mathbf{1}_{m_i < 0}\sigma(v \cdot g(-m)),
$$

at which point, we see that if $m_1, m_2 \geq 0$, this becomes $\frac{1}{2}\sigma(-v \cdot m)$, as it is if $m_1, m_2 \leq 0$. If $m_1 \geq 0$ and $m_2 \leq 0$, then you get $\lim_\lambda \mathbf{A}_1^\mu = -\frac{1}{2}\sigma(-v_1 m_1)$ and $\lim_\lambda \mathbf{A}_2^\mu = -\frac{1}{2}\sigma(-v_2 m_2)$ and likewise if $m_1 \leq 0$ and $m_2 \geq 0$.

Next consider the limit as $\lambda \to \infty$ of $\mathbf{A}_{ij}^\perp$ from (4.3), which we claim converges to 0. Write

$$
\mathbf{A}_{ij}^\perp = -\frac{1}{2}\mathbb{E}\Big[(X_\mu \cdot W_j^\perp)\mathbf{1}_{W_i \cdot X \geq 0}\sigma(-v \cdot g(WX_\mu))\Big] \tag{E.17}
$$
$$
- \frac{1}{2}\mathbb{E}\Big[(X_\mu \cdot W_j^\perp)\mathbf{1}_{W_i \cdot X_\mu < 0}\sigma(v \cdot g(-WX_\mu))\Big].
$$

These two terms are bounded similarly. The absolute value of the first of these is bounded by $(1/2)\mathbb{E}[|X_\mu \cdot W_j^\perp|]$ which is at most $(1/2)\sqrt{R_{jj}^\perp}\lambda^{-1/2}$ by (E.2). The second is analogously bounded. These evidently go to zero as $\lambda \to \infty$.

Finally, since $|\mathbf{B}_{ij}| \leq 1$, the quantity $g_{R_{ij}^\perp} = c_\delta \frac{v_i v_j}{\lambda}\mathbf{B}_{ij}$ evidently goes to zero as $\lambda \to \infty$.

*Remark* 4. The above argument used $m_i \neq 0$ for the limit of $\mathbf{A}_i^\mu$. If one considers the cases when $m_i = 0$, the limiting drifts still apply. For this, it suffices to show that if $m_i = 0$, then $\mathbf{A}_i^\mu$ converges to zero. Without loss of generality, suppose $m_1 = 0$ and consider

$$
\mathbf{A}_1 \cdot \mu = \mathbb{E}\big[Z_{1,\mu}\mathbf{1}_{Z_{1,\perp} \geq 0}\sigma(-v \cdot g(Z_{1,\perp}, m_2 Z_{2,\mu} + Z_{2,\perp}))\big].
$$

This is zero independently of $\lambda$ by independence of $Z_{1,\mu}$ from the other Gaussians in the expectation.

We next turn to classifying the fixed points of this limiting ODE system. Evidently, every fixed point must have $R_{ij}^\perp = 0$. Furthermore, if we let $u_i = v_i - m_i$, then

$$
\dot{u}_i = \begin{cases} -\frac{u_i}{2}\sigma(-v \cdot m) - \alpha u_i & m_1 m_2 > 0 \\ -\frac{u_i}{2}\sigma(-v_i m_i) - \alpha u_i & \text{else} \end{cases},
$$

and therefore every fixed point of the ODE system must have $u_i = 0$, which is to say $v_i = m_i$. Therefore, it suffices to characterize the fixed points in terms of $(v_1, v_2)$ as claimed. This reduces to

$$\begin{cases} v_i \sigma(-\|v\|^2) = 2\alpha v_i & v_1 v_2 > 0 \\ v_i \sigma(-v_i^2) = 2\alpha v_i & \text{else} \end{cases}.$$

Observe first that the point $(v_1, v_2) = (0, 0)$ is a fixed point of this system. If $(v_1, v_2) \neq 0$, then dividing out by $v_i$, the above reduces to

$$\begin{cases} \sigma(-\|v\|^2) = 2\alpha & v_1 v_2 > 0 \\ \sigma(-v_i^2) = 2\alpha & \text{else} \end{cases}.$$

Recalling that

$$C_\alpha = -\operatorname{logit}(2\alpha) = \log(1 - 2\alpha) - \log(2\alpha), \tag{E.18}$$

we obtain the claimed set of fixed points by inverting these equations (they only have a solution if $\alpha < 1/4$).

In order to study the stability of the various fixed points, notice first that the ODE system of Proposition 4.2 is a gradient system for the $\lambda = \infty$ population loss,

$$\Phi(v, m) = \frac{1}{2} \left( \log(1 + e^{-v \cdot g(m)}) + \log(1 + e^{v \cdot g(-m)}) \right) + \frac{\alpha}{2} \sum_{i=1,2} (v_i^2 + m_i^2 + R_{ii}^\perp).$$

Since it is a gradient system, with only the specified fixed points, the stability of a fixed point can be deduced by showing it is the minimizer of $\Phi$. In particular, the values of $\Phi$ at its critical points are given by $\Phi_0 = \log 2$ at $v_1 = v_2 = 0$, $\Phi_+ = \frac{1}{2}(\log 2 + \log(1 + e^{-C_\alpha})) + \alpha C_\alpha$ when $v_1 v_2 > 0$, and $\Phi_- = \log(1 + e^{-C_\alpha}) + 2\alpha C_\alpha$ when $v_1 v_2 < 0$. It is a simple calculus exercise to show that the smallest of these is $\Phi_0$ when $\alpha > 1/4$ and $\Phi_-$ when $\alpha < 1/4$.

To show that each of the other critical points are all unstable, one can find a direction along which the dynamical system is locally repelled from it. For instance, we will show that the ring of fixed points with $v_i = m_i$ and $R_{ij}^\perp = 0$ with $v_1 v_2 \leq 0$ is unstable, by showing a repelling direction arbitrarily close to the point $v_1 = -\sqrt{C_\alpha}$, $v_2 = 0$. If $v_1 = -\sqrt{C_\alpha}$ and $v_2 = \epsilon > 0$, then $\dot{v}_2$ there reduces to $\epsilon(\frac{\sigma(-\epsilon^2)}{2} - \alpha)$, and as long as $\alpha < 1/4$, there exists $\epsilon > 0$ such that $\sigma(-\epsilon^2) > 2\alpha$ so $\dot{v}_2 > 0$ for all $\epsilon$ small enough.

$\square$

### E.5 Rescaled effective dynamics around unstable fixed points

In this section, we consider scaling limits of the rescaled effective dynamics in their noiseless limit, where the rescaling is about the unstable set of fixed points given by the quarter circle $v_1^2 + v_2^2 = C_\alpha$ per item (2) of Proposition 4.2. In what follows, let $\delta_n = c_\delta/N$, and fix $(a_1, a_2) \in \mathbb{R}_+^2$ with $a_1^2 + a_2^2 = C_\alpha$, and let $\mathbf{u}_n$ be the variables of (4.2) with $v_i, m_i$ replaced by $\tilde{v}_i = \sqrt{N}(v_i - a_i)$ and $\tilde{m}_i = \sqrt{N}(m_i - a_i)$.

**Proof of Proposition 4.3.** We start by considering the drift process for these rescaled variables. Notice that the rescaling induces the transformation $\tilde{J}$ multiplying $J$ by $\sqrt{N}$ in its entries corresponding to $v_i, m_i$. The fact that the rescaled variables satisfy the conditions of Theorem 2.2 follows as in Lemma E.3 with the only distinction arising in the bound on (E.11), where previously we did not use the $\delta_n^2$ factor—in the new coordinates, the factor of $\sqrt{N}$ raised to the fourth power is cancelled out by $\delta_n^2$ as long as $\delta_n = O(1/N)$.

For the population drift of the new variables, if the variables $\tilde{v}_i, \tilde{m}_i$ are in a ball of radius $K$ in $\mathbb{R}^4$ (which we take to be our $E_K$), the signs of $m_i$ agree, and therefore

$$f_{\tilde{v}_i} = -\sqrt{N} f_{v_i} = -\sqrt{N} \frac{v_i}{2} \sigma(-v \cdot m) + \alpha \sqrt{N} m_i$$

$$f_{\tilde{m}_i} = -\sqrt{N} f_{m_i} = -\sqrt{N} \frac{m_i}{2} \sigma(-v \cdot m) + \alpha \sqrt{N} v_i.$$

We wish to claim that these expressions have consistent limits when $\tilde{v}_i, \tilde{m}_i$ are localized to $E_K$ for fixed $K$. notice that in $m_i = a_i + N^{-1/2}\tilde{m}_i$ and $v_i = a_i + N^{-1/2}\tilde{v}_i$, and using $\sum a_j^2 = C_\alpha$,

$$v \cdot m = C_\alpha + N^{-1/2} \sum_{j=1,2} a_j(\tilde{v}_j + \tilde{m}_j) + O(1/n) \,.$$

Now Taylor expanding the sigmoid function, and using the definition of $C_\alpha$, we get

$$\sigma(-v \cdot m) = \sigma(-C_\alpha) + (v \cdot m - C_\alpha)\sigma(-C_\alpha)(1 - \sigma(-C_\alpha)) + O(n^{-1})$$
$$= 2\alpha + N^{-1/2}a_j\Big( \sum_{j=1,2} (\tilde{v}_j + \tilde{m}_j)(2\alpha)(1 - 2\alpha) + O(n^{-1}) \,.$$

Plugging these into the earlier expressions for $f_{\tilde{v}_i}$, we see that

$$f_{\tilde{v}_i} = -\frac{N^{1/2}a_i + \tilde{m}_i}{2}\Big(2\alpha + \frac{1}{N^{1/2}}a_j \sum_{j=1,2} (\tilde{v}_j + \tilde{m}_j)(2\alpha)(1 - 2\alpha) + O\Big(\frac{1}{n}\Big)\Big) + \alpha(n^{1/2}a_i + \tilde{v}_i)$$

$$= -\alpha\tilde{m}_i + \alpha\tilde{v}_i - a_i(\alpha - 2\alpha^2) \sum_{j=1,2} a_j(\tilde{v}_j + \tilde{m}_j) + O(n^{-1/2}) \,.$$

Taking the limit as $n \to \infty$, this yields exactly the population drift claimed for the $\tilde{v}_i$ variable. The calculation for $f_{\tilde{m}_i}$ is analogous, and the equations for $R_{ij}^\perp$ are evidently unchanged by the transformation of $v_i, m_i$ to $\tilde{v}_i, \tilde{m}_i$. Furthermore, these variables are still linear so no corrector is introduced.

We now turn to computing the limiting diffusion matrix $\Sigma$ in the new variables $\tilde{v}_i, \tilde{m}_i$. We first use the following expression for the matrix $V$ when $\lambda = \infty$, by taking the $\lambda = \infty$ in (E.10).

$$V_{v_i,v_j} = \frac{m_i m_j}{4} \cdot \begin{cases} \sigma(-v \cdot m)^2 & m_1 m_2 > 0 \\ \sigma(-v_i m_i)\sigma(-v_j m_j) & \text{else} \end{cases},$$

$$V_{v_i,W_j} = \frac{m_i v_j}{4}\mu \cdot \begin{cases} \sigma(-v \cdot m)^2 & m_1 m_2 > 0 \\ \sigma(-v_i m_i)\sigma(-v_j m_j) & \text{else} \end{cases},$$

$$V_{W_i,W_j} = \frac{v_i v_j}{4}\mu^{\otimes 2} \cdot \begin{cases} \sigma(-v \cdot m)^2 & m_1 m_2 > 0 \\ \sigma(-v_i m_i)\sigma(-v_j m_j) & \text{else} \end{cases}.$$

Rewriting these in the coordinates $\tilde{v}$ and $\tilde{m}$, we see that in $E_K$,

$$V_{v_i,v_j} = \alpha^2 a_i a_j + O(n^{-1/2}) \,, \qquad V_{v_i,W_j} = \mu(\alpha^2 a_i a_j + O(n^{-1/2})) \,,$$

and

$$V_{W_i,W_j} = \mu^{\otimes 2}(\alpha^2 a_i a_j + O(n^{-1/2})) \,.$$

Now multiplying this on both sides by $\tilde{J}$, for the $\tilde{\mathbf{u}}_n$ variables, the two factors of $\sqrt{N}$ from $\tilde{J}$ cancel out with the choice of $\delta_n = 1/N$, and in the $n \to \infty$ limit, leave

$$\tilde{\Sigma}_{v_i v_j} = \tilde{\Sigma}_{m_i m_j} = \tilde{\Sigma}_{v_i m_j} = \alpha^2 a_i a_j \,,$$

as claimed. □

## F  Deferred proofs from Section 5

Fix two orthogonal vectors $\mu, \nu \in \mathbb{R}^N$ and recall the cross-entropy loss with penalty $p(v, W) = \frac{\alpha}{2}(\|v\|^2 + \|W\|^2)$. For the XOR GMM with SGD, the cross-entropy loss is given by

$$L(v, W) = -yv \cdot g(WX) + \log\big(1 + e^{v \cdot g(WX)}\big) + p(v, W) \tag{F.1}$$

where if the class label $y = 1$, then $X$ is a symmetric binary Gaussian mixture with means $\pm\mu$, and if $y = 0$, then $X$ is a symmetric Gaussian mixture with means $\pm\nu$. This has the same form as the loss for the 2-layer binary GMM, and we will find many similarities in the below between them. Indeed, the only difference is in the distribution of $X$ conditionally on the class label $y$ as described, and the fact that $v$ is now in $\mathbb{R}^4$ and $W = (W_i)_{i=1,\dots,4}$ is now a $4 \times N$ matrix. In what follows we take $n = 4N + 4$. As such, all the formulae of (E.3)– (E.10) also hold for the XOR GMM, but with the law of $(y, X)$ now understood differently.

*Remark* 5. In principle, we can take $W$ to be $k \times d$ and $v$ to be a $k$ vector, but $4$ is the first reasonable choice of $k$, as if $k < 4$ the network cannot express a good classifier. Taking $k$ to be larger than $4$ is interesting, and can in principle be handled by our methods–we leave this for future investigation. We could also have added a bias at each layer, however the Bayes classifier in this problem is an "X" centered at the origin so we can safely take the biases to be 0.

## F.1 Summary statistics and localizability

Recall the set of summary statistics $\mathbf{u}_n$ from (5.1). The next lemma shows that $\mathbf{u}_n$ form a good set of summary statistics.

**Lemma F.1.** *The distribution of $L((v, W))$ depends only on $\mathbf{u}_n$ from (5.1). In particular, we have that $\Phi(x) = \phi(\mathbf{u}_n)$ for some $\phi$. Furthermore, $\mathbf{u}_n$ satisfy the bounds in item (1) of Definition 2.1 if $E_K$ is the ball of radius $K$ in $\mathbb{R}^{4N+4}$.*

*Proof.* Let $X_w = \mathcal{N}(w, I/\lambda)$ for $w \in \{\mu, -\mu, \nu, -\nu\}$. Notice that the law of $L$ at a fixed point $(v, W) \in \mathbb{R}^{4+4N}$ can be written as

$$L((v, W)) \stackrel{d}{=} \begin{cases} -v \cdot g(WX_\mu) + \log(1 + e^{v \cdot g(WX_\mu)}) + p(v, W) & \text{w. prob. } 1/4 \\ -v \cdot g(WX_{-\mu}) + \log(1 + e^{v \cdot g(WX_{-\mu})}) + p(v, W) & \text{w. prob. } 1/4 \\ \log(1 + e^{v \cdot g(WX_\nu)}) + p(v, W) & \text{w. prob. } 1/4 \\ \log(1 + e^{v \cdot g(WX_{-\nu})}) + p(v, W) & \text{w. prob. } 1/4 \end{cases} \tag{F.2}$$

Next, notice that as a vector

$$WX_\iota = (m_i + Z_{i,\iota} m_i^\iota + Z_{i\perp})_{i=1,\dots,4} \qquad \text{for } \iota \in \{\mu, \nu\},$$

where $Z_{i,\iota}$ are i.i.d. $\mathcal{N}(0, \lambda^{-1})$ and $(Z_{i\perp})$ are jointly Gaussian with covariance matrix

$$\text{Cov}(Z_{i\perp}, Z_{j\perp}) = \lambda^{-1} R_{ij}^\perp.$$

Similarly, the law of $WX_{-\iota}$ depends only on $(m_i^\iota, R_{ij}^\perp)$. Finally,

$$p(v, W) = \tfrac{\alpha}{2} \sum_{i=1,\dots 4} \left( v_i^2 + R_{ii}^\perp \right).$$

Therefore, at a fixed point $(v, W)$ the law of $L(v, W)$ is only a function of $\mathbf{u}_n(v, W)$.

To see that the summary statistics satisfy the bounds of item (1) in Definition 2.1, note that the non-zero entries of $J = (\nabla u_\ell)_\ell$ are as follows.

$$\partial_{v_i} v_i = 1, \qquad \nabla_{W_i} m_i^\mu = \mu, \qquad \nabla_{W_i} m_i^\nu = \nu, \qquad \nabla_{W_i} R_{jk}^\perp = W_j^\perp \delta_{ij} + W_k^\perp \delta_{ik}, \tag{F.3}$$

where $\delta_{ij}$ is 1 if $i = j$ and 0 otherwise. For higher derivatives, we only have second derivatives in the $R_{jk}^\perp$ variables, each of which is given by a block diagonal matrix where only one block is non-zero and it is twice an identity matrix. Thus the operator norm of these second derivatives is 2. The third derivatives of all elements of $\mathbf{u}_n$ are zero. $\qquad \square$

In the following, let

$$\mathbf{A}_i = \mathbb{E}\left[ X \mathbf{1}_{W_i \cdot X \geq 0} \left( -y + \sigma(v \cdot g(WX)) \right) \right].$$

By the same reasoning as in Lemma F.1, if $w \in \{\mu, \nu, W_i, W_i^\perp\}$, then $w \cdot \mathbf{A}_i$ is only a function of $\mathbf{u}_n$. We then also have the conclusions of Lemma E.2 for $X$ distributed according to the XOR GMM by simply decomposing it into two mixtures, and we will therefore appeal to this lemma meaning its analogue for the XOR GMM.

**Lemma F.2.** *For $\delta = O(1/N)$ and any fixed $\lambda$, the 2-layer XOR GMM with observables $\mathbf{u}_n$ is $\delta_n$-localizable for $E_K$ being balls of radius $K$ about the origin in $\mathbb{R}^{22}$.*

*Proof.* The condition on $\mathbf{u}_n$ was satisfied per Lemma F.1. Recalling $\nabla \Phi$ from (E.6)–(E.7), one can verify that the norm of each of the four terms in $\nabla \Phi$ is individually bounded, using the Cauchy–Schwarz inequality together with the bound of Lemma E.2 on $\|\mathbf{A}_i\|$, naturally adapted to XOR. The remaining estimates are also analogous to the proof of Lemma E.3 with the analogue of Lemma E.2 applied. $\qquad \square$

## F.2 Effective dynamics for the XOR GMM

For a point $(v, W) \in \mathbb{R}^{4+4N}$, let

$$\mathbf{A}_i^{\mu} = \mu \cdot \mathbf{A}_i \,, \qquad \mathbf{A}_i^{\nu} = \nu \cdot \mathbf{A}_i \,, \qquad \mathbf{A}_{ij}^{\perp} = W_j^{\perp} \cdot \mathbf{A}_i \,.$$

Furthermore, let

$$\mathbf{B}_{ij} = \mathbb{E}\big[\mathbf{1}_{W_i \cdot X \geq 0} \mathbf{1}_{W_j \cdot X \geq 0}\big(-y + \sigma(v \cdot g(WX))\big)^2\big] \,.$$

**Proposition F.1.** *Let $\mathbf{u}_n$ be as in (5.1) and fix any $\lambda > 0$ and $\delta_n = c_\delta/N$. Then $\mathbf{u}_n(t)$ converges to the solution of the ODE system $\dot{\mathbf{u}}_t = -\mathbf{f}(\mathbf{u}_t) + \mathbf{g}(\mathbf{u}_t)$, initialized from $\lim_n (\mathbf{u}_n)_* \mu_n$ with*

$$f_{v_i} = m_i^{\mu} \mathbf{A}_i^{\mu}(\mathbf{u}) + m_i^{\nu} \mathbf{A}_i^{\nu}(\mathbf{u}) + \mathbf{A}_{ii}^{\perp}(\mathbf{u}) + \alpha v_i \,, \qquad f_{m_i^{\mu}} = v_i \mathbf{A}_i^{\mu} + \alpha m_i^{\mu} \,,$$

$$f_{R_{ij}^{\perp}} = v_i \mathbf{A}_{ij}^{\perp}(\mathbf{u}) + v_j \mathbf{A}_{ji}^{\perp}(\mathbf{u}) + 2\alpha R_{ij}^{\perp} \,, \qquad f_{m_i^{\nu}} = v_i \mathbf{A}_i^{\nu} + \alpha m_i^{\nu} \,.$$

*and correctors $g_{v_i} = g_{m_i^{\mu}} = g_{m_i^{\nu}} = 0$, and $g_{R_{ij}^{\perp}} = c_\delta \frac{v_i v_j}{\lambda} \mathbf{B}_{ij}$ for $1 \leq i \leq j \leq 4$.*

*Proof.* The convergence of the population drift to $\mathbf{f}$ from Proposition 4.1 follows by taking the inner products of $\nabla L$ from (E.6) with the rows of $J$ from (F.3), and noticing that $\mathbf{A}_i^{\mu}$ is exactly $\mathbf{A}_i \cdot \mu$, $\mathbf{A}_i^{\nu}$ is exactly $\nu \cdot \mathbf{A}_i$, and $\mathbf{A}_{ij}^{\perp}$ is exactly $\mathbf{A}_i \cdot W_j^{\perp}$.

We next consider the population correctors. The fact that $g_{v_i} = g_{m_i^{\mu}} = g_{m_i^{\nu}} = 0$ follows from the fact that the Hessians of $v_i, m_i^{\mu}, m_i^{\nu}$ are zero. For the corrector $g_{R_{ij}^{\perp}}$ for $1 \leq i \leq j \leq 4$, the relevant entries of $V$ are those corresponding to $W_i^{\perp}$ and $W_j^{\perp}$. For ease of notation, in what follows let $\pi = \sigma(v \cdot g(WX))$.

Similar to the calculation of (E.12),

$$\delta_n \mathcal{L}_n R_{ij}^{\perp} = \frac{c_\delta}{N} v_i v_j \Big( \mathbb{E}\big[\|X^{\perp}\|^2 \mathbf{1}_{W_i \cdot X \geq 0} \mathbf{1}_{W_j \cdot X \geq 0} (\pi - y)^2 \big]$$
$$- \big\langle \mathbf{A}_i - \mathbf{A}_i^{\mu} \mu - \mathbf{A}_i^{\nu} \nu, \mathbf{A}_j - \mathbf{A}_j^{\mu} \mu - \mathbf{A}_j^{\nu} \nu \big\rangle \Big) \,.$$

By the same arguments on the concentration of the norm of Gaussian vectors as used in the binary GMM case, then we deduce from this that

$$g_{R_{ij}^{\perp}} = \frac{c_\delta v_i v_j}{\lambda} \mathbb{E}\big[\mathbf{1}_{W_i \cdot X \geq 0} \mathbf{1}_{W_j \cdot X \geq 0}(-y + \pi)^2\big] = \frac{c_\delta v_i v_j}{\lambda} \mathbf{B}_{ij} \,.$$

Finally, let us establish that the limiting diffusion matrix is all-zero whenever $\delta_n = o(1)$. This follows exactly as it did in the proof of Proposition 4.1. $\qquad \square$

## F.3 Small noise limit of the effective dynamics

The aim of this section is to establish the following small-noise $\lambda \to \infty$ limit of the effective dynamics ODE of Proposition F.1. This will again be quite similar to the analogous proofs for the binary GMM in Section E, and when these similarities are clear we will omit details.

**Proposition F.2.** *In the $\lambda \to \infty$ limit, the ODE from Proposition F.1 converges to*

$$\dot{v}_i = \frac{m_i^{\mu}}{4}\Big(\mathbf{1}_{m_i^{\mu}>0}\sigma(-v \cdot g(m^{\mu})) - \mathbf{1}_{m_i^{\mu}<0}\sigma(-v \cdot g(-m^{\mu}))\Big)$$
$$- \frac{m_i^{\nu}}{4}\Big(\mathbf{1}_{m_i^{\nu}>0}\sigma(v \cdot g(m^{\nu})) - \mathbf{1}_{m_i^{\nu}<0}\sigma(v \cdot g(-m^{\nu}))\Big) - \alpha v_i \,,$$

$$\dot{m}_i^{\mu} = \frac{v_i}{4}\Big(\mathbf{1}_{m_i^{\mu}>0}\sigma(-v \cdot g(m^{\mu})) - \mathbf{1}_{m_i^{\mu}<0}\sigma(-v \cdot g(-m^{\mu}))\Big) - \alpha m_i^{\mu} \,,$$

$$\dot{m}_i^{\nu} = -\frac{v_i}{4}\Big(\mathbf{1}_{m_i^{\nu}>0}\sigma(-v \cdot g(m^{\nu})) - \mathbf{1}_{m_i^{\nu}<0}\sigma(-v \cdot g(-m^{\nu}))\Big) - \alpha m_i^{\nu} \,,$$

*and $\dot{R}_{ij}^{\perp} = -2\alpha R_{ij}^{\perp}$ for $1 \leq i \leq j \leq 4$.*

*Proof.* Let us begin with convergence of $\mathbf{A}_i^\mu$. We claim that it converges to

$$\lim_{\lambda \to \infty} \mathbf{A}_i^\mu = -\frac{1}{4}\mathbf{1}_{m_i^\mu > 0}\sigma(-v \cdot g(m^\mu)) - \frac{1}{4}\mathbf{1}_{m_i^\mu < 0}\sigma(v \cdot g(-m)).$$

In order to see this, expand

$$\mathbf{A}_i = \frac{1}{4}\mathbb{E}\big[-X_\mu \mathbf{1}_{W_i \cdot X_\mu \geq 0}(\sigma(-v \cdot g(WX_\mu)))\big] - \frac{1}{4}\mathbb{E}\big[X_{-\mu}\mathbf{1}_{W_i \cdot X_{-\mu} \geq 0}(\sigma(-v \cdot g(WX_{-\mu})))\big]$$
$$+ \frac{1}{4}\mathbb{E}\big[X_\nu \mathbf{1}_{W_i \cdot X_\nu \geq 0}(\sigma(v \cdot g(WX_\nu)))\big] + \frac{1}{4}\mathbb{E}\big[X_{-\nu}\mathbf{1}_{W_i \cdot X_{-\nu} \geq 0}(\sigma(v \cdot g(WX_{-\nu})))\big].$$

The point will be that when taking the inner product with $\mu$, the first two terms here contribute to the limit and the latter two vanish, while when taking the inner product with $\nu$, the first two terms vanish in the $\lambda \to \infty$ limit while the latter two contribute.

Consider e.g., the first of the four terms above, and inner product with $\mu$. In this case, consider

$$\mathbb{E}\big[(X_\mu \cdot \mu)\mathbf{1}_{W_i \cdot X_\mu \geq 0}\sigma(-v \cdot g(WX_\mu))\big] - \mathbf{1}_{m_i^\mu \geq 0}\sigma(-v \cdot g(m^\mu)),$$

which is precisely the quantity that was exactly shown to go to zero as $\lambda \to \infty$ in (E.16). To see that the third and fourth terms above go to zero when taking their inner product with $\mu$, observe that they become

$$\big|\mathbb{E}\big[(X_\nu \cdot \mu)\mathbf{1}_{W_i \cdot X_\nu \geq 0}\sigma(v \cdot g(WX_\nu))\big]\big| \leq \mathbb{E}[|X_\nu \cdot \mu|],$$

which by orthogonality of $\mu$ and $\nu$ is at most $\lambda^{-1/2}$ by the reasoning of Lemma E.2, therefore vanishing as $\lambda \to \infty$. Together with its analogue for $X_{-\nu}$, this implies the claim for the convergence of $\mathbf{A}_i^\mu$, as well as its analogous limit of $\mathbf{A}_i^\nu$.

We next consider the limit as $\lambda \to \infty$ of $\mathbf{A}_{ij}^\perp$, which we claim goes to $0$. Using the expansion of $\mathbf{A}_i$ from earlier in this proof, we can consider $\mathbf{A}_{ij}^\perp = \mathbf{A}_i \cdot W_j^\perp$ as four terms having the form of the terms in (E.17), which were there showed to go to zero as $\lambda \to \infty$. Since $W_j^\perp$ here is orthogonal both to $\mu$ and $\nu$, the same proof applies.

Finally, in order to see that the limit as $\lambda \to \infty$ of $g_{R_{ij}^\perp} = c_\delta \frac{v_i v_j}{\lambda} \mathbf{B}_{ij}$ is zero, which follows from the fact that $|\mathbf{B}_{ij}| \leq 1$. $\qquad \square$

**Proposition F.3.** *The fixed points of the ODE system of Proposition F.2 are classified as follows. If $\alpha > 1/8$, then the only fixpoint is at $\mathbf{u}_n = \mathbf{0}$.*

*If $0 < \alpha < 1/8$, then let $(I_0, I_\mu^+, I_\mu^-, I_\nu^+, I_\nu^-)$ be any disjoint (possibly empty) subsets whose union is $\{1, ..., 4\}$. Each such partition fully dictates a connected component of fixpoints for that dynamial system. Corresponding to that tuple $(I_0, I_\mu^+, I_\mu^-, I_\nu^+, I_\nu^-)$, the connected component of fixpoints has $R_{ij}^\perp = 0$ for all $i, j$, and*

1. *$m_i^\mu = m_i^\nu = v_i = 0$ for $i \in I_0$,*

2. *$m_i^\mu = v_i > 0$ such that $\sum_{i \in I_\mu^+} v_i^2 = logit(-4\alpha)$ and $m_i^\nu = 0$ for all $i \in I_\mu^+$,*

3. *$-m_i^\mu = v_i > 0$ such that $\sum_{i \in I_\mu^-} v_i^2 = logit(-4\alpha)$ and $m_i^\nu = 0$ for all $i \in I_\mu^-$,*

4. *$m_i^\nu = v_i < 0$ such that $\sum_{i \in I_\nu^+} v_i^2 = logit(-4\alpha)$ and $m_i^\mu = 0$ for all $i \in I_\nu^+$,*

5. *$-m_i^\nu = v_i < 0$ such that $\sum_{i \in I_\nu^-} v_i^2 = logit(-4\alpha)$ and $m_i^\mu = 0$ for all $i \in I_\nu^-$.*

*There are $39$ connected components of fixed points. Of these, there are $4! = 24$ many that are stable, corresponding to the possible permutations in which each of $I_\mu^+, I_\mu^-, I_\nu^+, I_\nu^-$ are singletons.*

*Proof.* Evidently, any fixed point must have $R_{ij}^\perp = 0$ for all $i, j$. Furthermore, the point $v_i = m_i^\mu = m_i^\nu = 0$ for $i = 1, ..., 4$ evidently forms a fixed point of the system. Now suppose there is some fixed point with $v_i = 0$ for some $i$; in that case, it must be that $m_i^\mu = 0$ and $m_i^\nu = 0$. Therefore, we can select a subset $I_0$ of $\{1, ..., 4\}$ such that $v_i = m_i^\mu = m_i^\nu$ for $i \in I_0$.

For any such choice of $I_0$, consider next, $i \notin I_0$. We first claim that if $v_i > 0$ at a fixed point, then $m_i^\mu \in \{\pm v_i\}$ and $m_i^\nu = 0$, whereas if $v_i < 0$ then $m_i^\nu \in \{\pm v_i\}$ and $m_i^\mu = 0$. To see this, notice that at any fixed point,

$$4\alpha m_i^\mu = v_i \Big( \mathbf{1}_{m_i^\mu \geq 0} \sigma(-v \cdot g(m^\mu)) - \mathbf{1}_{m_i^\mu < 0} \sigma(-v \cdot g(-m^\mu)) \Big),$$

$$4\alpha m_i^\nu = -v_i \Big( \mathbf{1}_{m_i^\nu \geq 0} \sigma(-v \cdot g(m^\nu)) - \mathbf{1}_{m_i^\nu < 0} \sigma(-v \cdot g(-m^\nu)) \Big).$$

Since $\sigma$ is non-negative, if $v_i > 0$, the sign of the right-hand side of the first equation is the same as the sign of $m_i^\mu$ so it can have a non-zero solution, while the sign of the right-hand side of the second equation is the opposite of the sign of $m_i^\nu$, so any such fixed point must have $m_i^\nu = 0$. To see that $m_i^\mu = \pm v_i$ at such a fixed point, now set $m_i^\nu = 0$ and take the fixed point equations for $v_i$ and $m_i^\mu$, dividing one by $v_i$ and the other by $m_i^\mu$ to see that

$$4\alpha \frac{v_i}{m_i^\mu} = 4\alpha \frac{m_i^\mu}{v_i}, \qquad \text{or} \qquad v_i^2 = (m_i^\mu)^2,$$

as claimed. The fixed points having $v_i < 0$ are solved symmetrically.

Our classification now reduces to understanding the possible values taken by $(v_1, ..., v_4)$ given their signs (when non-zero). Fix a partition $(I_0, I_\mu^+, I_\mu^-, I_\nu^+, I_\nu^-)$ of $\{1, ..., 4\}$ and consider the set of fixed points having $m_i^\mu = m_i^\nu = v_i = 0$ for $i \in I_0$, $m_i^\mu = v_i > 0$ on $I_\mu^+$ and so on as designated by Proposition F.3; by the above any fixed point is of this form. It remains to check that the values of $v_i$ on each of these sets are as described by the proposition.

In order to see this, fix e.g., $i \in I_\mu^+$. Then, $m_i^\mu = v_i$ and $m_i^\nu = 0$, and so the fixed point equations reduce to

$$4\alpha v_i = v_i \sigma(-v \cdot g(m^\mu)), \qquad \text{or} \qquad 4\alpha = \sigma\Big( - \sum_{j \in I_\mu^+} v_j^2 \Big),$$

since the only coordinates where $g(m^\mu)$ will be non-zero are $j \in I_\mu^+$, where $m_j^\mu = v_j$. Inverting the sigmoid function, this implies exactly the claimed $\sum_{j \in I_\mu^+} v_j^2 = \text{logit}(-4\alpha)$. The cases of $I_\mu^-, I_\nu^+, I_\nu^-$ are analogous, concluding the proof.

Let us now count the number of connected components of fixed points. We first notice that the fixed point at $(0, ..., 0)$ is disconnected from all others. Fixed points corresponding to some $(I_0, ..., I_\nu^-)$ are part of the same connected component of fixed points if one goes from one to the other by moving an element of $I_\iota^\eta$ (for some $\iota \in \{\mu, \nu\}$ and $\eta \in \{\pm\}$) to $I_0$ without making $I_\iota^\eta$ empty, or by moving an element of $I_0$ to a non-empty $I_\iota^\eta$.

We turn now to studying the stability of these various sets of fixed points. Observe that in the $\lambda \to \infty$ limit, the dynamical system of Proposition F.2 is a gradient system for the population loss

$$\Phi = \frac{1}{4} \Big( \log(1 + e^{-v \cdot g(m^\mu)}) + \cdots + \log(1 + e^{-v \cdot g(-m^\nu)}) \Big)$$
$$+ \frac{\alpha}{2} \sum_i (v_i^2 + (m_i^\mu)^2 + (m_i^\nu)^2 + R_{ii}^\perp).$$

At a fixed point (which necessarily has $v_i = m_i$, $R_{ii}^\perp = 0$, and is characterized by the partition of $\{1, ..., 4\}$ into $I_\mu^+, I_\mu^-, I_\nu^+, I_\nu^-$, this reduces to

$$\Phi = \frac{1}{4} \Big( \log(1 + e^{-\sum_{i \in I_\mu^+} v_i^2}) + \cdots + \log(1 + e^{-\sum_{i \in I_\nu^-} v_i^2}) \Big) + \alpha \sum_i v_i^2$$

At this point, noticing that $\sum_{i \in I_\mu^+} v_i^2$ is equal to $C_\alpha = -\text{logit}(4\alpha)$ if $I_\mu^+$ is non-empty and $0$ if it is empty, and similarly for $I_\mu^-, I_\nu^+, I_\nu^-$, this turns into a simple optimization problem over the number of non-empty $I_\mu^+, I_\mu^-, I_\nu^+, I_\nu^-$. Just as in the binary GMM case, it becomes evident that when $\alpha > 1/8$, this is minimized at $v_i = 0$ for all $i$ (i.e., they are all empty and $I_0 = \{1, ..., 4\}$, whereas when $\alpha < 1/8$ the above is minimized when every one of $I_\mu^+, I_\mu^-, I_\nu^+, I_\nu^-$ are all non-empty. This yields the global minima of $\Phi$ in these coordinates, and ensures the fixed points we claimed were stable are indeed stable.

To show the instability of any other connected set of fixed points, the reasoning goes just as in the binary GMM case: consider a small perturbation of the specified critical region in the direction of the stable fixed points and it can be seen by examining the drifts directly, that the dynamical system has a repelling direction. □

### F.4 ³⁄₃₂-probability of ballistic convergence to an optimal classifier

Let us now reason that the ballistic effective dynamics of Proposition F.2 is such that under an uninformative Gaussian initialization, the probability of being in a basin of attraction of one of the 24 stable fixed points is $3/32$. Begin by noticing that if the first layer weights are initialized as $W_i \sim \mathcal{N}(0, I_N/N)$ independently for $i = 1, ..., 4$ and the second layer weights $v_i$ are independent standard Gaussians, then the projection onto the coordinate system $(v_i, m_i^\mu, m_i^\nu, R_{ij})$ is given by

$$\lim(\mathbf{u}_n)_* \mu_n = \mathcal{N}(0, 1)^{\otimes 4} \otimes \delta_0^{\otimes 4} \otimes \delta_0^{\otimes 4} \otimes \delta_{I_4}$$

For the $m_i^\mu, m_i^\nu$ variables one should understand these $\delta_0$ Dirac masses as a $\frac{1}{2}$-$\frac{1}{2}$ mixture of $\delta_{0^-}$ and $\delta_{0^+}$. Under the flow of Proposition F.2, if $v_i(0)$ is positive, then $m_i^\nu$ stays fixed at zero, and if $m_i^\mu(0) = 0^-$ then $m_i^\mu$ becomes negative infinitesimally quickly, whereas if $m_i^\mu(0) = 0^+$ then it becomes positive infinitesimally quickly. At any rate, the sign of $v_i$ never changes to negative from such an initialization, and similarly if $v_i(0)$ is negative, the sign of $v_i$ will never change to positive. As such, in order to have a chance at being in the basin of attraction of one of the stable fixed points outlined in Proposition F.3, it must be the case that two of $(v_i(0))_i$ have positive sign and two of them have negative sign; evidently this has probability $\binom{4}{2}/2^4 = 3/8$.

Given that two of $v_i(0)$ are positive, and two of them are negative—say without loss of generality that $i = 1, 2$ are the coordinates in which it is positive, and $i = 3, 4$ are the coordinates in which it is negative—then the dynamical system for $(v_1, v_2, m_1^\mu, m_2^\mu)$ is exactly the ballistic limit of the two-layer GMM studied in Section E, for which we found that the probability of converging to a good classifier is $1/2$. Similarly, the dynamical system for $(v_2, v_4, m_3^\nu, m_4^\nu)$ independently gives a further probability $1/2$ of converging to *its* good classifier. Together, these yield a probability of $3/32$ of converging to one of the 4! many optimal classifiers for the XOR GMM.

### F.5 Diffusive limit on critical submanifolds

We now consider scaling limits of the rescaled effective dynamics in their noiseless limit, where the rescaling is about the unstable set of fixed points given by the product of two quarter circles where $I_\mu^+ = \{1, 2\}$ and $I_\nu^+ = \{3, 4\}$. In what follows, fix $(a_{1,\mu}, a_{2,\mu}) \in \mathbb{R}_+^2$ with $a_{1,\mu}^2 + a_{2,\mu}^2 = C_\alpha$, and $a_{3,\nu}^2 + a_{4,\nu}^2 = C_\alpha$, and let $\mathbf{u}_n$ be the variables of (4.2) with $v_i, m_i^\mu, m_i^\nu$ replaced by

$$\tilde{v}_i = \begin{cases} \sqrt{N}(v_i - a_{i,\mu}) & i = 1, 2 \\ -\sqrt{N}(v_i - a_{i,\nu}) & i = 3, 4 \end{cases}$$

and

$$\tilde{m}_i^\mu = \begin{cases} \sqrt{N}(m_i^\mu - a_{i,\mu}) & i = 1, 2 \\ 0 & i = 3, 4 \end{cases}, \qquad \tilde{m}_i^\nu = \begin{cases} 0 & i = 1, 2 \\ \sqrt{N}(m_i^\nu - a_{i,\nu}) & i = 3, 4 \end{cases}.$$

By the choices of $\tilde{m}_i^\mu = 0$ and $\tilde{m}_i^\nu = 0$, we mean that we formally mean that we remove those variables from $\tilde{\mathbf{u}}_n$, and for us now $E_K$ will be the ball of radius $K$ in the other coordinates, and the point $\{0\}$ for $(\tilde{m}_i^\mu)_{i=3,4}$ and $(\tilde{m}_i^\nu)_{i=1,2}$.

**Proof of Proposition 5.1** The fact that the rescaled variables $\tilde{\mathbf{u}}_n$ satisfy the conditions of Theorem 2.2 follows as in Lemma F.2 with the only distinction arising in the bound on (E.11), where previously we did not use the $\delta_n^2$ factor, but is still satisfied using $\delta_n = O(1/n)$.

We next consider the population drift of the new variables $\tilde{v}_i, \tilde{m}_i^\mu$ and $\tilde{m}_i^\nu$. If we take these variables to be in $E_K$, and recall the population drifts etc. in the $\lambda = \infty$ setting from Proposition F.2, for $i = 1, 2$, we have $f_{\tilde{v}_i}$ is the $n \to \infty$ limit of

$$\sqrt{N} \frac{m_i^\mu}{4} \sigma(-v \cdot g(m^\mu)) - \sqrt{N} \alpha v_i$$

If we then use the expansion

$$v \cdot g(m^\mu) = C_\alpha + N^{-1/2} \sum_{j=1,2} a_{j,\mu}(\tilde{v}_j + \tilde{m}_j^\mu) + O(1/n)$$

from which we obtain

$$\sigma(-v \cdot g(m^\mu)) = \sigma(-C_\alpha) + \frac{1}{\sqrt{N}}\Big( \sum_{j=1,2} a_{j,\mu}(\tilde{v}_j + \tilde{m}_j^\mu)\Big)(4\alpha)(1 - 4\alpha) + O(\tfrac{1}{n})$$

Plugging these in, and taking the $n \to \infty$ limit we find that for $i = 1, 2$,

$$f_{\tilde{v}_i} = \alpha(\tilde{v}_i - \tilde{m}_i^\mu) - a_{i,\mu}(\alpha - 4\alpha^2) \sum_{k=1,2} a_{k,\mu}(\tilde{v}_k + \tilde{m}_k^\mu) \,.$$

By a similar reasoning, for $i = 3, 4$, we have

$$f_{\tilde{v}_i} = \alpha(\tilde{v}_i - \tilde{m}_i^\nu) - a_{i,\nu}(\alpha - 4\alpha^2) \sum_{k=3,4} a_{k,\nu}(\tilde{v}_k + \tilde{m}_k^\nu) \,.$$

The claimed equations for $f_{\tilde{m}_i^\mu}$ when $i = 1, 2$ and $f_{\tilde{m}_i^\nu}$ when $i = 3, 4$ hold by analogous reasoning, and the equations for $f_{R_{ij}^\perp}$ are evidently unaffected by the change of variables to $\tilde{\mathbf{u}}_n$. Regarding the population correctors, they are also unaffected (all zero) since the variables that were changed in $\tilde{\mathbf{u}}_n$ are all linear.

It remains to compute the volatility matrix in the coordinates $v_i, \tilde{m}_i^\mu, \tilde{m}_i^\nu$. We first use the following expression for the matrix $V$ when $\lambda = \infty$, by taking $\lambda = \infty$ in (E.10). If $i, j \in \{1, 2\}$, then

$$V_{v_i, v_j} = \begin{cases} \frac{3}{16} m_i^\mu m_j^\mu \sigma(-v \cdot m^\mu)^2 & i, j \in \{1, 2\} \\ \frac{3}{16} m_i^\nu m_j^\nu \sigma(v \cdot m^\nu)^2 & i, j \in \{3, 4\} \end{cases}$$

and if $i \in \{1, 2\}$ and $j \in \{3, 4\}$, then

$$V_{v_i, v_j} = -\frac{1}{16} m_i^\mu m_j^\nu \sigma(-v \cdot m^\mu)\sigma(v \cdot m^\nu)$$

When considering $\Sigma_{v_i, v_j}$ we multiply this by $N$ coming from $\tilde{J}$ and $\tilde{J}^T$, but also multiply by $\delta = 1/N$, so that taking the limit as $n \to \infty$, we get

$$\tilde{\Sigma}_{v_i, v_j} = \begin{cases} 3\alpha^2 a_{i,\mu} a_{j,\mu} & i, j \in \{1, 2\} \\ 3\alpha^2 a_{i,\nu} a_{j,\nu} & i, j \in \{3, 4\} \\ -3\alpha^2 a_{i,\mu} a_{j,\nu} & i \in \{1, 2\}, j \in \{3, 4\} \end{cases} .$$

By a similar reasoning, if $i, j \in \{1, 2\}$, then

$$V_{v_i, W_j} \cdot \mu = \frac{3}{16} v_j m_i^\mu \sigma(-v \cdot m^\mu)^2 \qquad i, j \in \{1, 2\}$$

$$V_{v_i, W_j} \cdot \nu = \frac{3}{16} v_j m_i^\nu \sigma(v \cdot m^\nu)^2 \qquad i, j \in \{3, 4\}$$

and if $i \in \{1, 2\}$ and $j \in \{3, 4\}$, then

$$V_{v_i, W_j} \cdot \nu = -\frac{1}{16} v_j m_i^\mu \sigma(-v \cdot m^\mu)\sigma(v \cdot m^\nu) \,.$$

Taking the limit as $n \to \infty$, we again recover the claimed limiting diffusion matrix, and similar calculations yield the same for $\Sigma_{\tilde{m}_i^\mu, \tilde{m}_j^\mu}$, $\Sigma_{\tilde{m}_i^\nu, \tilde{m}_j^\nu}$ and $\Sigma_{\tilde{m}_i^\mu, \tilde{m}_j^\nu}$, concluding the proof. $\qquad\square$