# OpenReview forum: "High-dimensional limit theorems for SGD: Effective dynamics and critical scaling"
_NeurIPS.cc/2022/Conference — NeurIPS 2022 Accept_

### Official Review · Reviewer_ju2h · 2022-07-06

**Rating:** 8
**Confidence:** 4
**Soundness:** 4 excellent
**Presentation:** 3 good
**Contribution:** 3 good

**Summary:**

This paper studies rigorous approximations of SGD dynamics. Assuming an SGD algorithm of the form
$$ X_{n+1} = X_n - \delta \nabla L(X_n, Y_n), $$
and a set of summary statistics $u: \mathbb R^p \to \mathbb R^k$ for fixed $k$, the authors show that the dynamics of $u$ converge in probability
to a stochastic differential equation of the form
$$ du = (f(u) +  g(u))dt + \Sigma(u)dW_t.$$

The authors also provide a phenomenology of the three terms in the RHS, wherein $f(u)$ corresponds to the gradient flow term, $g(u)$ is a deterministic correction term, and $\Sigma$ encapsulates the variance of each SGD step. Depending on the chosen stepsize $\delta$, those terms can be either influential or negligible in the associated SDE.

The remainder of the article is devoted to several examples. First, the authors study the use of SGD for matrix/tensor PCA:
$$ Y = \lambda x^{\otimes k} + W, $$
where $W$ is an i.i.d Gaussian $k$-tensor. With careful renormalizations of the dynamics, they map the SGD problem to an Ornstein-Uhlenbeck process, and show that the threshold for stability of this OU process matches the already known thresholds for hardness in terms of $\lambda$.

The last two examples refer to Gaussian mixture classification; one is a simple binary task, while the other is the so-called XOR Gaussian mixture, which is not amenable to a simple linear separator. In both cases, the authors consider a two-layer neural network, and provide a full set of sufficient statistics to describe the dynamics. Those dynamics are characterized by a complex structure of fixed points (resp. 4 and 625 connected components), some of those being unstable; the authors then demonstrate how "zooming in" on the dynamics helps characterize the behaviour around those fixed points.



**Questions:**

- Can you provide additional explanations regarding the 3/32 probability of ending up at a stable fixed point in the XOR mixture ?
- What would happen if, like in many real-world networks, the number of hidden units (and therefore, the number of sufficient statistics) starts to diverge ?

**Limitations:**

The limitations are adequately addressed.

**Strengths And Weaknesses:**

This paper is the culmination of several previous works studying rigorous approximations of SGD dynamics (e.g. Tan and Vershynin, Ben Arous et al., Veiga et al. ...); it provides a very general result encompassing all previous works. The hypotheses are quite reasonable, and less restrictive than most usual restrictions (no uniform Lipschitz condition on $u$, only fourth moment conditions on $L$).

The provided examples illustrate quite well both the power and the limitations of this approach: on the one hand, the main theorem is applicable to a wide array of learning problems, and yields a very precise characterization as an SDE. On the other hand, those SDEs are highly non-trivial to study, even in the ballistic phase without any noise. Hence, there is still a lot of work to extract interesting insights from this theorem on a new learning task.

The main weakness of this paper is its terseness. The main theorem is introduced without giving much intuition about it, while the gist of the proof (disregarding the important aspects of martingale bounding) is based on a Taylor expansion of $u(X_{n+1})$ in powers of $\delta$. This would give a much needed intuitive interpretation to the correction term and the diffusion matrix.

---

> ### Author Response · Authors · 2022-08-01
> **Response to referee ju2h**
>
> We sincerely thank the referee for their detailed comments and helpful suggestions.
>
> Regarding terseness: we have added an Appendix B which discusses in detail each of the items in Definition 2.1. We hope this clarifies the role each of these conditions play in Theorem 2.2 and why they should hold for a wide class of high-dimensional problems. For intuition on the proof of Theorem 2.2, and in particular the suggested Taylor expansion heuristic, due to space considerations, we will expand on this in a full version and thank the referee for the suggestion.
>
> We have added a short proof of the 3/32 probability of ballistic convergence to an optimal classifier to Appendix F.4. We apologize for our oversight in not having earlier included this and thank the referee for pointing that out.
>
> The question of what happens when the number of summary statistics also diverges is a very interesting one that unfortunately doesn't fit directly into our framework; we leave this to future investigation.

---

### Official Review · Reviewer_hQ5R · 2022-07-06

**Rating:** 7
**Confidence:** 3
**Soundness:** 3 good
**Presentation:** 3 good
**Contribution:** 3 good

**Summary:**

This work investigates the high-dimensional limit for the evolution of summary statistics under one-pass stochastic gradient descent (SGD).

Given some data, a loss function and a learning rate, the one-pass SGD dynamics is defined by evaluating the gradient *at a single data sample* and performing a gradient step in this direction. In the so-called *classical* limit where the dimensions of the problem are fixed and the learning rate is taken to zero, one-pass SGD is known to converge to the gradient flow in the population risk. This work characterises the corrections to the gradient flow dynamics arising when the dimensions of the problem (data dimension, number of parameters) scale relatively to the learning rate.

Its main theoretical contribution is to map the one-pass SGD dynamics into a set of tractable low-dimensional stochastic differential equations (SDE) for the evolution of the summary statistics (i.e. low-dimensional functions of interest of the parameters, such as their covariance and correlation with a ground truth).

Explicit examples such as planted matrix and tensor PCA and Gaussian mixture classification are discussed in detail, where it is shown how the SDE simplifies in different regimes associated to different choices of scalings for the dimensions.

**Questions:**

-**[Q1]**: In L137, "fixed dimension" means both fixed $d_{n}$ and $p_{n}$?

-**[Q2]**: In [49] it was shown that for teacher-student two-layer NNs by letting the hidden-layer width grow faster than the data dimension, one can go from a regime of imperfect to perfect learning (corresponding to the cross-over between a critical to sub-critical regime in the author's vocabulary). Can a similar conclusion be drawn from Theorem 2.2 (for instance by letting $p_{n}$ grows faster than $d_{n}$), or does it depend on the setting?

-**[Q3]**: In the examples discussed, the stochastic term $\Sigma$ is only present around a fixed point of the drift part of the dynamics. Is there anything in the theorem preventing us from having $\Sigma\neq 0$ outside a fixed point?

-**[Q4]**: What the authors mean by "converging to a unstable fixed point" (L268-L273, L316-L317)? I understand that the dynamics can get trapped in a neighbourhood of an unstable fixed points for some time if initialised close to it, but how can it converge to a unstable fixed point?

**Limitations:**

An inevitable limitation of the level of generality of this work is that it does not provide a constructive way to choose the summary statistics for a given problem of interest. This could be briefly commented in the text as well.

**Strengths And Weaknesses:**

The investigation of one-pass SGD through the lenses of its summary statistics has a long history, which to my best knowledge dates back to the seminal work of Saad & Solla [41, 42], and was followed by an intense research activity since the mid-90s [BS, RB, CC]. These early works were inspired by ideas from statistical physics of disordered systems, where reducing the study of a high-dimensional random system to the study of low-dimensional equations for quantities that concentrate (known in that field as "order parameters") is quite natural. These early works have focused in the particular setting of two-layer neural networks (NNs) in a teacher-student setting (i.e. when the training data is generated by a two-layer NN itself) and in the critical scaling (adopting the author's terminology), but since have been enlarged to other data models, such as correlated Gaussians [20], Gaussian mixture classification [37], and to other scaling regimes [49]. I stress that because although [41] has been published in this very own venue (NeurIPS'95), closely related ideas have recently resurfaced in the mathematical machine learning literature, but missed this connection (e.g. [TV, 52]) - likely because they were independently rediscovered.

In this work, the authors take an important step forward by collecting and extending these specific settings into a general abstract framework for mapping the one-pass SGD dynamics of a probabilistic learning task into a SDE for the summary statistics. In particular, the characterisation of the stochastic correction in terms of a simple diffusion is an important contribution that goes beyond the literature above, and that allows to study how the dynamics escape from unstable fixed points.

However, although most of the classic literature discussed above is cited in this manuscript, the connections are poorly acknowledged on the level of the discussion. For instance:

- The trade-off between the "population drift" $f$ and the "population corrector" $g$ in the critical scaling already appeared in [41, 42] in the specific context of teacher-student two-layer networks. There, it was shown that the presence of $g$ gives rise to a fixed point that prevents the dynamics to reach perfect learning at linear time scales.

- The cross-over between the "critical" and "sub-critical" regimes (when the correction $g=0$) with the relative scaling between the data dimension $d_{n}$, the number of parameters $p_{n}$ and the learning rate $\delta_{n}$ was recently discussed in [49]. This should be reflected in the discussion in L63-L67 and L141-L144.

- The SNR scaling for the existence transition of good classifiers for one-pass SGD on the Gaussian Mixture XOR classification task with two-layer NNs was discussed in [37].

In my opinion, the manuscript is also well-written and the conceptual thread is easy to follow. However, I believe the technical part in Section 2 and in particular Definition 2.1 is hard to parse for the general NeurIPS audience. It would be nice if the authors could add some intuition on what each of the conditions 1-3 mean, and maybe the simplest concrete example the author's can think of a localising and a non-localising sequence to help the reader develop an intuition.

To summarise, the strengths and weaknesses of the manuscript are:

**Strengths**: The manuscript is well written and easy to follow. The theoretical contribution is a significant addition to the literature, setting a fairly general framework to understand the behaviour of one-pass SGD through tractable low-dimensional equations. The examples given are pertinent and useful to understand the theory.

**Weaknesses**: The discussion is not well placed in the relevant literature. Some technical parts of the manuscript are hard to parse.


**References** (numbered refs. are from the bibliography in the paper)

[BS] M. Biehl and H. Schwarze, "Learning by on-line gradient descent", Journal of Physics A: Mathematical
and General, vol. 28, no. 3, pp. 643–656, feb 1995.

[RB] P. Riegler and M. Biehl. "On-line backpropagation in two-layered neural networks". Journal of
Physics A: Mathematical and General, 28(20), 1995.

[CC] M. Copelli and N. Caticha, “On-line learning in the committee machine,” Journal of Physics A: Math-
ematical and General, vol. 28, no. 6, pp. 1615–1625, mar 1995.

[TV] Y.S. Tan, R. Vershynin, "Online Stochastic Gradient Descent with Arbitrary Initialization Solves Non-smooth, Non-convex Phase Retrieval", arXiv: 1910.12837 [stat.ML]

---

> ### Author Response · Authors · 2022-08-01
> **Response to referee hQ5R**
>
> We sincerely thank the referee for their detailed comments and helpful suggestions.
>
> We have expanded on the relation of our work to the works of Saad and Solla, as well as Veiga et. al. in the places the referee requested. We thank the referee for explaining the relations of our work to those two important works, and apologize for not having realized this connection earlier.
>
> Regarding the difficulty of parsing the localizability condition: we have added two extended remarks in what is now Appendix B that discuss in detail each of the items in Definition 2.1. We hope these help clarify the role each of these play and why they should hold for a wide class of high-dimensional problems.
>
> To respond to the questions the referee asked:
>
> (1) Yes, by "fixed dimensions" here we mean regimes in which both the dimension of the data and the parameter space are fixed constants, and the number of data points tends to infinity.
>
> (2) It is a very interesting question to probe over-parametrized regimes where $p_n \gg d_n$ and we leave this to future investigation.
>
> (3) In general, nothing precludes this possibility since Theorem 2.2 is stated in rather broad generality.
> In particular, one does not need to apply it to a region near a fixed point in all coordinates to see a stochastic term.
> One could even take a combination of variables being tracked that have different scalings, some of which are initialized essentially near their ``optimal" value and some moving ballistically. In this case, one might wish to rescale the first kind of variables to understand fluctuations while the remaining variables move ballistically. We will provide examples of this last scenario in a full version.
>
> (4) We thank the referee for giving us the opportunity to discuss this very important point.
> As we demonstrate in our GMM-based examples, the diffusive scaling limits near fixed points of the ballistic dynamics correspond to truly degenerate diffusions. In particular, the volatility matrix may go to zero as one approaches an unstable fixed point. Such a degenerate diffusion can then have a behavior similar to geometric Brownian motion (see, e.g., the behavior of $\tilde v_i$ in Proposition 4.3 as $a_i\to 0$) which depending on initialization can be trapped indefinitely near a repelling zero of its drift.

---

> > ### Comment · Reviewer_hQ5R · 2022-08-03
> > **Follow-up**
> >
> > I thank the authors for the time spent answering my questions and concerns. I appreciate it. I might need some time to fully digest the replies to all the other reviewers. Meanwhile, to make the best of the discussion period let me follow up in a reply.
> >
> > > (3) In general, nothing precludes this possibility since Theorem 2.2 is stated in rather broad generality. In particular, one does not need to apply it to a region near a fixed point in all coordinates to see a stochastic term. One could even take a combination of variables being tracked that have different scalings, some of which are initialized essentially near their ``optimal" value and some moving ballistically. In this case, one might wish to rescale the first kind of variables to understand fluctuations while the remaining variables move ballistically. We will provide examples of this last scenario in a full version.
> >
> > Thank you for clarifying this. I see how one could construct such an example by combining summary statistics at different scalings (and I look forward for the explicit example in the full version).
> >
> > However, if instead of looking at the summary statistics we focus on the evolution of the error (mse for tensor-PCA, population risk for GMM classification, etc). Can the authors envisage a situation where the stochastic contribution is non-zero outside a fixed point of the drift? I know this is sort of an open ended question, but I would like to hear the author's intuition on this point.

---

> > > ### Author Response · Authors · 2022-08-04
> > > **Re: follow-up**
> > >
> > > In the specific examples we have looked at, for the evolution of the error, as well as other summary statistics, we expect the stochastic contribution to only be non-negligible in neighborhoods of the fixed points of the ballistic dynamics.
> > >
> > > In order for this not to be the case for some statistic $u$, it would have to be that the \emph{variance} of the SGD increment is especially large in the specific direction of $\nabla u$ (namely, of an order $\delta_n^{-1}$ larger than it is in most other directions).

---

> > > > ### Comment · Reviewer_hQ5R · 2022-08-08
> > > > **Re: Re: follow-up**
> > > >
> > > > I thank the authors for detailing more this aspect. That's clear.
> > > >
> > > > Given the authors' openness to the reviewers suggestions - which I think will help making the work more accessible to the NeurIPS community - as well as their effort for clarifying some points during the discussion period, I am happy to raise my score towards a stronger accept.

---

### Official Review · Reviewer_uVoh · 2022-07-27

**Rating:** 7
**Confidence:** 3
**Soundness:** 3 good
**Presentation:** 2 fair
**Contribution:** 3 good

**Summary:**

This paper investigates the high-dimensional limit of SGD with constant step-size. The authors obtain an ODE/SDE type of limiting dynamics for the trajectory of a given finite set of summary statistics when the dimension of the data and the parameters go to infinity. The limiting dynamics is shown to be matching with the gradient flow for the population loss when the step-size is below the sub-critical scaling regime. When the step-size is in the critical scaling regime, it is shown that the limiting dynamics might possibly get a ballistic correction term and a diffusive term. In addition, the authors also develop a separate diffusive (SDE) limit for the rescaled summary statistics around a fixed-point of the unscaled ballistic dynamics in order to investigate the microscopic behavior of the dynamics around those fixed-points. The authors also demonstrate the applicability of their main theorem in several examples including tensor spike model, and classification with two-layer neural networks of binary and XOR-type Gaussian mixtures.

**Questions:**

I would suggest the authors to include a notation section or a table in the main text or in the appendix. The introduction can be expanded more to motivate the subject better and can be divided into explicit subsections of prior works and contribution in order to clarifiy the limitations of the prior works and the advantages of your results better. It could also be helpful to mention the prior work and compare with your results in sections 3-5. Figures could be explained better either in the main text or in the captions. A summary illustration visualizing different limiting dynamics under different scaling regimes can included but the page limit might not allow that.

I would also like to ask the following questions to the authors:
1. What can you say about implicit bias/regularization of SGD in the critical scaling regime or in the microscopic regime with your main theorem? Can the population correction term be written as the gradient of another function which might imply that SGD in critical regime minimizes a new loss function with an additive bias term?
2. What can your result tell about the over-parameterized regime where $p_n/d_n \gg 1$?

**Limitations:**

Authors can discuss limitations of their results and future work in more detail. They mention the limiting dynamics of time-de4pendent rescaling as a future work in footnote 2.

**Strengths And Weaknesses:**

Strengths:
1. The main result seems to apply to a general class of loss functions and data distributions compared to prior works.
2. The theoretical claims are supported by experiments and explicit analysis of several benchmark models.

Weaknesses:
1. Presentation and the structure of the paper make it harder to follow.
2. Contribution over the prior works is not very clear.

Overall, I find this paper quite interesting and in good quality in terms of technical analysis. However, the material could be presented better for clarity.

---

> ### Author Response · Authors · 2022-08-01
> **Response to referee uVoh**
>
> We sincerely thank the referee for their detailed comments and helpful suggestions, especially given the time constraint of an emergency review.
>
> We have added a notation section in Appendix A as suggested by the referee.
>
> With respect to the organizational style: we apologize for not following the style perhaps more common in machine learning conferences. Given the space constraints and the audience of this conference we have omitted a background discussion on the importance of stochastic gradient descent to the Stats/ML community. We have attempted to include an extensive historical discussion of prior work  going back to the 1950s in the first four paragraphs of our paper to position our work as compared to the classical literature on asymptotic theory and the more recent works in high-dimensions and discuss our main contributions in the next six paragraphs respectively. We have added signposts to this discussion as requested by the referee.
>
> Furthermore, in the ``Our contributions" section, we have better clarified the connection between our results of research and the most closely related line of works, namely the work of Saad and Solla and the very recent works of Veiga et. al. (both in the case of teacher-student networks).
>
> We have expanded the caption to Figure 1 to make it more descriptive. A host of simulations will be made available in the full version of this paper; as anticipated by the reviewer, space limitations preclude us from including these here.
>
> To respond to the questions the referee raised:
>
> (1) We show here that even in simple (though not overparametrized) examples like the XOR one, SGD may not have implicit regularization and in high-dimensional regimes, converges to classifiers with sub-optimal generalization error. Studying this in overparametrized regimes would be of interest (see also item below) and we thank the referee for this suggestion.
>
> With regards to the second part of the question, neither the population drift, nor the population corrector are necessarily of gradient type. In particular, even the population dynamics are a possibly non-linear function of a higher dimensional gradient flow, and transformations of this type can lead to non-gradient systems.
>
> (2) The scaling relation between $p_n$ and $d_n$ in Theorem 2.2 is not constrained, so the theorem applies in generality. It would be interesting to consider the limiting dynamics obtained by overparametrized versions of our examples for instance and we leave this to future investigation.

---

> > ### Comment · Reviewer_hQ5R · 2022-08-03
> > **Clarification**
> >
> > I completely share reviewer's *uVoh* view on the presentation style, and therefore I would also endorse her/his request for expanding the introduction and motivation in a future version (I am well aware of the space constraints at this stage).
> >
> > > *"With regards to the second part of the question, neither the population drift, nor the population corrector are necessarily of gradient type. In particular, even the population dynamics are a possibly non-linear function of a higher dimensional gradient flow, and transformations of this type can lead to non-gradient systems. "*
> >
> > Can the authors develop further on this interesting point? In the discussion in L145-147, the author's remark that the population drift $f$" corresponds to *"evolution under gradient descent on the population loss $\Phi$"*. Why it is not necessarily of gradient type?

---

> > > ### Author Response · Authors · 2022-08-04
> > > **Re: clarification**
> > >
> > > Thanks for the question. While the population dynamics for $X_t$ is of gradient type by definition, it can be that the function $\mathbf{f}$ is not the gradient of any scalar function $F:\mathbb R^k \to \mathbb R$ because of the non-linear projection from parameter space to the summary statistics.
> > >
> > > For example, in the case of matrix PCA with $u_1 = m$ and $u_2 = r_\perp^2$, the population dynamics of the summary statistics is given by
> > > $$\dot u_1 = 2 \lambda k u_1 - 2 k u_1^3  - 2k u_1 u_2\quad  \mbox{and}\quad \dot u_2 = - 4k u_1^2 u_2 - 4k u_2^2.$$
> > > It can be checked by taking antiderivatives that the right-hand sides are not the $(u_1, u_2)$ derivatives of any scalar function on $\mathbb R^2$. In the special case where the population loss $\Phi$ is a function $\phi$ of the summary statistics, and the summary statistics are all linear functions, the population dynamics will indeed be of gradient type $\mathbf{f} = \nabla \phi$.
> > >
> > > In the course of writing this, we realized that we might have misunderstood referee uVoh's original question. In particular, one might ask whether the effective drift is the projection of a corrected gradient dynamics, namely if there is a function $F$ such that the effective drift for the summary statistics $\mathbf{u}$ is of the form $\langle DF,D\mathbf{u}\rangle$. Even though this holds for the population drift where $\mathbf{f} = \langle D\Phi,D\mathbf{u}\rangle$, it is not necessarily the case for the full drift because the correction term corresponds to a second-order (as opposed to first-order) differential operator on $\mathbf{u}$.

---

> > > > ### Comment · Reviewer_hQ5R · 2022-08-08
> > > > **Re: Re: clarification**
> > > >
> > > > That's clear. I thank the authors for the clarification.

---

> > ### Comment · Reviewer_uVoh · 2022-08-08
> > **Re: Response to referee uVoh**
> >
> > Dear authors,
> >
> > I really appreciate the time you took to answer my questions and clarify certain points. I was also confused by your answer to my question about the gradient-like form of the drift and the bias term as reviewer hQ5R pointed out. But the authors clarified this issue quite well for me. I would also like to thank reviewer hQ5R for addressing this misunderstanding. Based on the technical strength and the significant contribution of this paper as well as the willingness of the authors to revise the presentation style and the organization, I am willing to increase my rating to  '7:accept'.

---

### Meta-Review · Area_Chair_TzQf · 2022-08-26

**Recommendation:** Accept
**Confidence:** Certain

**Metareview:**

The paper is quite interesting and rigorous, with intriguing conclusions. The rebuttal also addressed all the major concerns -- mostly technical clarity. I congratulate the authors for the nice work and recommend an acceptance for the paper.

**Award:**

Yes

---

### Decision · Program_Chairs · 2022-09-14

Accept